# Myelin regulatory factor (MYRF) is a critical early regulator of retinal pigment epithelial development

**Michelle L. Brinkmeier**[1,2], **Su Qing Wang**[1], **Hannah A. Pittman**[1], **Leonard Y. Cheung**[2,3], **Lev Prasov**[1,2]*

1 Department of Ophthalmology and Visual Sciences, University of Michigan, Ann Arbor, Michigan, United States of America, 2 Department of Human Genetics, University of Michigan, Ann Arbor, Michigan, United States of America, 3 Department of Physiology and Biophysics, State University of New York at Stony Brook, Stony Brook, New York, United States of America

* lprasov@umich.edu

## Abstract

Myelin regulatory factor (Myrf) is a critical transcription factor in early retinal and retinal pigment epithelial development, and human variants in *MYRF* are a cause for nanophthalmos. Single cell RNA sequencing (scRNAseq) was performed on *Myrf* conditional knockout mice (*Rx > Cre Myrf*<sup>fl/fl</sup>) at 3 developmental timepoints. *Myrf* was expressed specifically in the RPE, and expression was abrogated in *Rx > Cre Myrf*<sup>fl/fl</sup> eyes. scRNAseq analysis revealed a loss of RPE cells at all timepoints resulting from cell death. GO-term analysis in the RPE revealed downregulation of melanogenesis and anatomic structure morphogenesis pathways, which were supported by electron microscopy and histologic analysis. Novel structural target genes including *Ermn* and *Upk3b*, along with macular degeneration and inherited retinal disease genes were identified as downregulated, and a strong upregulation of TGFß/BMP signaling and effectors was observed. Regulon analysis placed *Myrf* downstream or parallel to *Pax6* and *Mitf* and upstream of *Sox10* in RPE differentiation. Together, these results suggest a strong role for MYRF in the RPE maturation by regulating melanogenesis, cell survival, and cell structure, in part acting through suppression of TGFß signaling and activation of *Sox10*.

## Author summary

*MYRF* is a gene that regulates the development and function of the retinal pigmented epithelium (RPE), which play an important role in maintaining photoreceptor structure and function. Genetic variants in patients have been implicated in eye size disorders, particularly causing a small, but structurally normal eye. We have utilized a molecular technique, single cell RNA sequencing, to investigate how loss of *Myrf* specifically in the RPE in a mouse model impacts

**Data availability statement:** Single cell RNA sequencing data is deposited in GEO (Accession number: GSE269597, GSE269598, GSE269599). Primary data is provided for all the experiments. All additional relevant data can be found within the article and its supplementary information and primary data for images and analysis are deposited in DeepBlue repository (https://deepblue.lib.umich.edu/data/concern/data_sets/tm70mw254/).

**Funding:** This work was supported with grants from the National Eye Institute (NEI, https://www.nei.nih.gov/, K08-EY032098, K12-EY022299 to LP), the Bright Focus Foundation (https://www.brightfocus.org/, M2022011N to LP), the Research to Prevent Blindness Career Development Award (https://www.rpbusa.org/, to LP), the Knights Templar Eye Foundation Career Starter Award and Competitive Renewal (https://www.ktef.org/, to LP), the E. Matilda Ziegler Foundation for the Blind (http://emz-foundation.com/, to LP), and the Glaucoma Research Foundation (https://glaucoma.org/, to LP). The NEI P30 core grant (P30EY007003) helped support histology, imaging, and molecular biology. The funders had no role in study design, data collection and analysis, decision to publish, or preparation of the manuscript.

**Competing interests:** The authors have declared that no competing interests exist.

downstream gene expression at three developmental timepoints and used this information to define the role of *Myrf* in development. Our work identified key cytoskeletal structural genes specific to the RPE, *Ermn* and *Upk3b*, and a gene important for the cell survival, *Sox10*, as critical targets of *Myrf*. In addition, we have identified and confirmed that the TGFbeta signaling pathway is dysregulated when *Myrf* is lost during development. This pathway is particularly relevant in RPE health and eye growth. Our electron microscopy and histologic analyses also confirm a defect in RPE structure and function. We place *MYRF* within a hierarchy of genes involved in RPE development and introduce novel candidate genes for further study as retinal degeneration and nanophthalmos candidate genes.

## Introduction

The retinal pigmented epithelium, RPE, is a single layer of polarized epithelium that develops between the neural retina and the choroidal vasculature and sclera [1]. The RPE provides support critical for maintenance of the outer segments of the photoreceptors and regulates transport of nutrients and waste between the choroidal vasculature and the neural retina. The RPE is formed from the dorsal outer layer of the optic vesicle, which is derived from the anterior neural plate under the direction of eye field transcription factors including *Lhx2, Nr2e1, Otx2, Rax, Six3, Six6, Tbx3,* and *Pax6* [2,3]. *Pax6* together with *Pax2* and *Otx2* regulate the expression of the earliest marker of the pigmented RPE, *Mitf.* In turn, *Mitf* is a master regulator of RPE development. Loss of *Mitf* leads to transdifferentiation of RPE to neural retina and ectopic expression of *Mitf* activates genes involved in melanogenesis [4,5].

Many signaling pathways are important in the differentiation, maintenance, and function of the RPE (reviewed in [3,6,7]). TGFß, BMP, and WNT ligands in the periocular mesenchyme and surface ectoderm surrounding the lens drive the RPE fate [8]. Inactivation of WNT signaling in the early forming RPE inhibits the differentiation of RPE, likely through ß-*catenin* activation of *Otx2* and *Mitf* [9,10]. *Bmp7* is expressed throughout the RPE and loss of *Bmp7* in a mouse model leads to failure of eye development [11]. *Bmp4* is expressed in the dorsal region of the developing eye and loss of *Bmp4* leads to transdifferentiation of neural retina to RPE [12]. In addition, *Bmp2* in the RPE has been shown to be a negative regulator of eye growth in chick studies [13,14]. In mice, postnatal eye growth is regulated by repression of *Lrp2* and *Bmp2* by *Srebp2* [15]. The Hippo pathway also regulates RPE development and specification. Loss of YAP in early optic cup stages of eye development results in a transdifferentiation of RPE to retina [16]. Conditional deletion of YAP with the *Rx > cre* results in hypopigmentation of the RPE, a folding and thinning of the retinal layer, transdifferentiation of the RPE into retina, and altered expression of key RPE transcription factors. YAP is also crucial in maintaining the polarity of the RPE which impacts the survival of the retinal progenitor cells [16].

Maintenance of structural integrity and cellular polarity in RPE is critical to its function and the survival of overlying photoreceptors (reviewed in [17] and [18]). Adherens and tight junctions formed between the epithelial cells create a barrier between the vasculature of the choroid and the retinal cells while allowing for the transport of metabolites, nutrients, and the phagocytosis and disposal of photoreceptor outer segments. In addition, the basement membrane of the RPE cells sits on Bruch's membrane which provides support to the RPE cells and strengthens the choroid/retinal barrier. Epithelial to mesenchymal transition (EMT) of the RPE cells occurs in many ocular disease states including proliferative vitreoretinopathy (PVR) and age-related macular degeneration (AMD). TGFß signaling, specifically through TGFB2, has been identified as a driving mechanism of EMT leading to compromised RPE structure and disease progression [19].

*Myelin regulatory factor* (*MYRF*) is a membrane-associated transcription factor critical for development and maintenance of myelination. Pathogenic *MYRF* variants have been identified in families with autosomal dominantly inherited isolated nanophthalmos, a condition characterized by a small, structurally normal eye, as well as a feature of ocular cardiac urogenital syndrome [20–27]. We previously showed that biallelic loss of *Myrf* in the developing eye results in dysfunction of the retinal pigmented epithelium and subsequent retinal degeneration [28,29]. However, the precise site of action and function of *Myrf* in eye development has been debated [29,30]. To elucidate the role of *Myrf* in eye development and place it in the hierarchy of RPE differentiation, we performed single cell RNAseq (scRNAseq) on early eye field conditional knockout *Myrf* mouse model, *Rx > cre Myrf*<sup>fl/fl</sup>, at multiple developmental time points. Our findings place *Myrf* within the hierarchy of transcription factors known to regulate RPE development and help define the molecular mechanisms and signaling pathways driving RPE dysfunction in the absence of *Myrf*.

## Results

### MYRF is specifically expressed in the retinal pigment epithelium

While dominant mutations in MYRF have been definitively shown to cause nanophthalmos in humans, the specific cellular drivers of phenotype remain a subject of debate. There is speculation that the ciliary margin/ciliary zonules [30], RPE [28], and retina [31] may all be implicated. To better define specific cell-type of action for MYRF, we systemically evaluated the expression pattern of *Myrf* in mouse using RNAscope *in situ* hybridization. We found that *Myrf* mRNA is predominantly expressed in the RPE throughout retinal development (Fig 1A-1D), with much lower or no expression in the ciliary body, lens (Fig 1K and 1L) and retina. Conditional deletion of *Myrf* in the early eye field using an *Rx > Cre* transgene leads to loss of *Myrf* mRNA signal in the RPE, supporting the specificity of the probes (Fig 1E-1H). Optic nerve expression of *Myrf* is consistent with its known role in regulation of myelination, and is not altered by *Rx > cre* deletion, given that developing oligodendrocytes or progenitors do not express *Rx* (Fig 1I-1J). These data are consistent with prior qRT-PCR data from the human eye [28] and supported by existing scRNAseq datasets (Fig 2). To confirm that the phenotype previously reported is driven by a loss of *Myrf* in the RPE, we took advantage of an inducible RPE-*Tyrcre>ERT2* to delete *Myrf* in the RPE during embryogenesis. Timed pregnant females were injected with tamoxifen at e11.5, e12.5, and e13.5 and eyes were analyzed at e18.5. Loss of *Myrf* using the *RPE-Tyrcre>ERT2* recapitulated the phenotypes observed in the *Rx > cre Myrf*<sup>fl/fl</sup> model including loss of pigmentation, and reduced TMEM98 expression (S1 Fig). Together, these data support that the RPE is the primary site of function for MYRF within the eye.

### Loss of MYRF alters RPE cell number during development

To define the mechanism by which loss of MYRF leads to retinal degeneration [28] and eye size defects, we performed scRNAseq using the 10X genomics platform to characterize the gene expression changes within the RPE and neighboring cell types in *Rx > cre Myrf*<sup>fl/fl</sup> mice as compared to *Myrf*<sup>fl/fl</sup> controls using 3 specific developmental timepoints: embryonic day (e)13.5 – before histologic phenotypes are apparent in the *Rx > cre Myrf*<sup>fl/fl</sup> eyes [28]; e15.5 – the first time at which overt RPE pigmentation phenotypes appear; and postnatal day (P)0 – a time at which secondary changes

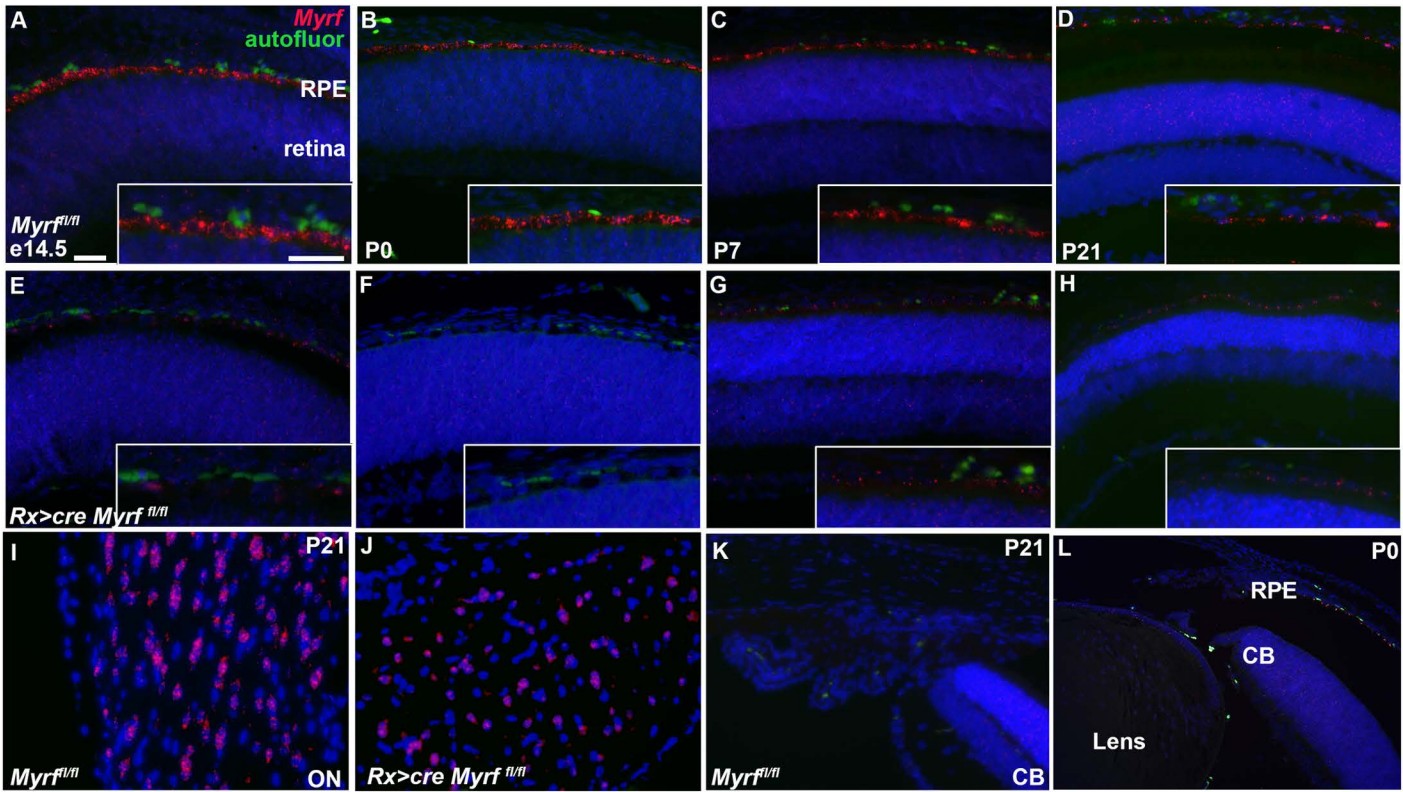

**Fig 1. *Myrf* is specifically expressed in the RPE during mouse development.** (A-H) *Myrf* transcripts in the RPE were detected by RNAscope during mouse embryonic development through postnatal day 21 in *Myrf*<sup>fl/fl</sup> controls (A-D) and absent in *Rx>cre Myrf*<sup>fl/fl</sup> mutants (E-H). (I-J) Expression of *Myrf* in the optic nerve (ON) serves as a positive control and was not abrogated with *Rx>Cre* deletion. Expression of *Myrf* is not detected in the ciliary body (CB, K) or lens (L). Green cells represent auto fluorescent choroidal red blood cells. Scale bar, 50uM.

may be apparent in neighboring cell types. We collected 6039, 13017, and 14305 cells for e13.5, e15.5, and P0 wild-type, respectively and 6734, 13880, and 12222, cells for e13.5, e15.5, and P0 mutant timepoints respectively. The median genes per cell were comparable in wild-type and mutant samples (2826, 2985, and 1915 for e13.5, e15.5, and P0 *Myrf*<sup>fl/fl</sup> and 2957, 2862, and 2179 for respective *Rx>cre Myrf*<sup>fl/fl</sup> eyes). After quality control filtering for dead cells, doublets, and poor-quality cells and combination and integration of the data, we conducted unsupervised clustering and were able to define 28 clusters, with 17 showing distinct gene expression patterns (Figs 2A and S2). The 7 retinal progenitor clusters, 3 photoreceptor clusters, 3 retinal ganglion cell clusters, and 2 periocular mesenchyme clusters that showed similar gene expression patterns and position on the UMAP dimensionality reduction plot were thus combined into the respective RPC, PhRec, RGC, and POM clusters for further analysis. We used *Pax3* expression, a specific marker of melanocytes and not RPE [32], to delineate RPE and Melanocyte clusters (S3 Fig). Analysis of *Myrf* transcripts within the clusters again suggested specific expression within the RPE during development, and loss of expression in *Rx>cre Myrf*<sup>fl/fl</sup> eye cups as expected (Fig 2B). All cell types were represented in both the *Rx>cre Myrf*<sup>fl/fl</sup> eyes and controls, but there were population differences with a reduction in RPE cell proportions in the mutant mice across all three developmental time points (5% vs. 3% at e13.5, 4% vs. 2% at e15.5, 5% vs. 3% at P0) (Fig 2C and 2D). ßIII-tubulin (TUJ1) staining, which marks neuronal lineages, is restricted to the retina in *Rx>cre Myrf*<sup>fl/fl</sup> mutants, suggesting that RPE does not transdifferentiation into neural retina when *Myrf* is lost (S4 Fig).

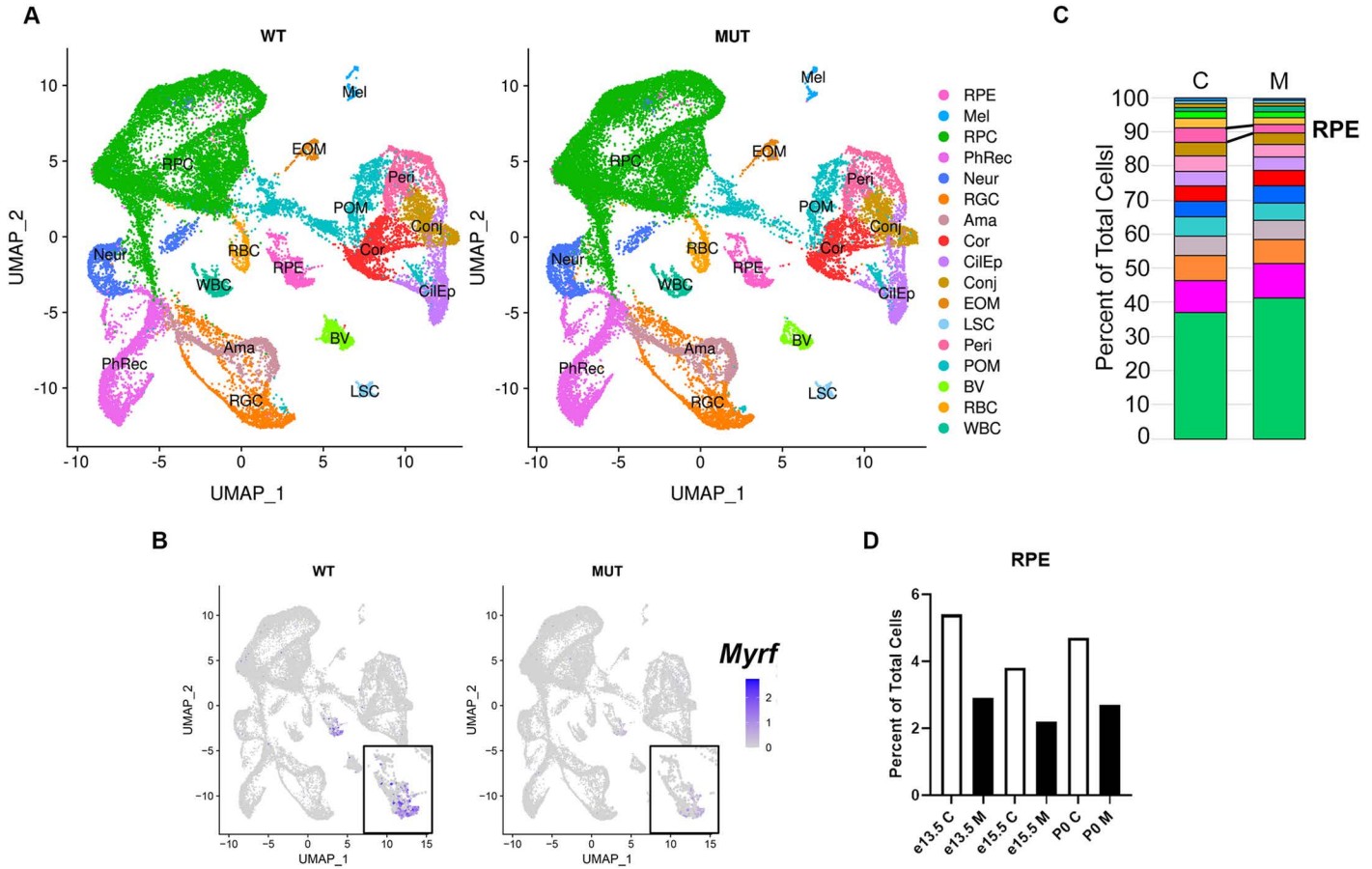

**Fig 2. scRNAseq analysis of *Rx>cre Myrf^fl/fl* mice shows decrease in the RPE cell population.** (A) Combined UMAP plots from e13.5, e15.5, and P0 *Myrf^fl/fl* (n=3 pooled for each time point) and *Rx>cre Myrf^fl/fl* (n=3 pooled for each time point) samples showing cell populations and cluster identification. (B) UMAP plot showing expression of *Myrf* specific to the RPE cluster in the control group and absent in the mutant group. (C) Stacked bar chart showing population distribution of clusters as a percentage of the total number of cells (control, C; mutant, M). The RPE cluster is highlighted as having a smaller cell proportion in the mutant. (D) Bar charts show the decrease in cell number within the RPE mutant (M) cluster across each time point.

## MYRF loss leads to RPE cell death without altered cell cycle dynamics

To further define the source of population level differences, we systematically examined cell cycle dynamics and cell death among the RPE population. Cleaved caspase-3 staining revealed an increase in cell death at e15.5, while there was no cell death in *Myrf^fl/fl* or *Rx>cre Myrf^+/fl* controls at postnatal (P0, P3) timepoints (S5 Fig). To further quantify cell death, we systematically evaluated apoptosis in the RPE at e14.5 using Terminal deoxynucleotidyl transferase dUTP nick end labeling (TUNEL), a timepoint immediately prior to the observable gross phenotype of RPE depigmentation. Co-staining with OTX2 was used to definitively mark RPE cells and identify retinal progenitors (Fig 3A-3C). Apoptotic cells were identified in the RPE of both *Rx>cre Myrf^+/fl* and *Rx>cre Myrf^fl/fl* mice, but very rarely in *Myrf^fl/fl* controls (insets, Fig 3Ai-3Ci). The percentage of TUNEL positive RPE cells was found to be significantly different in *Myrf^fl/fl* mice and Rx>cre *Myrf^fl/fl* eyes (0.6±0.7% vs. 11±9%, p=0.0122) (Fig 3D, and S1 Data). To exclude altered cellular proliferation and cell cycle dynamics in RPE accounting for population differences, we conducted cell cycle analysis in Seurat, which demonstrated similar levels of G1, S, and G2/M phase cells across the genotypes at each time point (S6 Fig). Subclustering of RPE followed by pseudotime trajectory analysis using Monocle3 reveals a modest decrease in cells with higher pseudotime values at

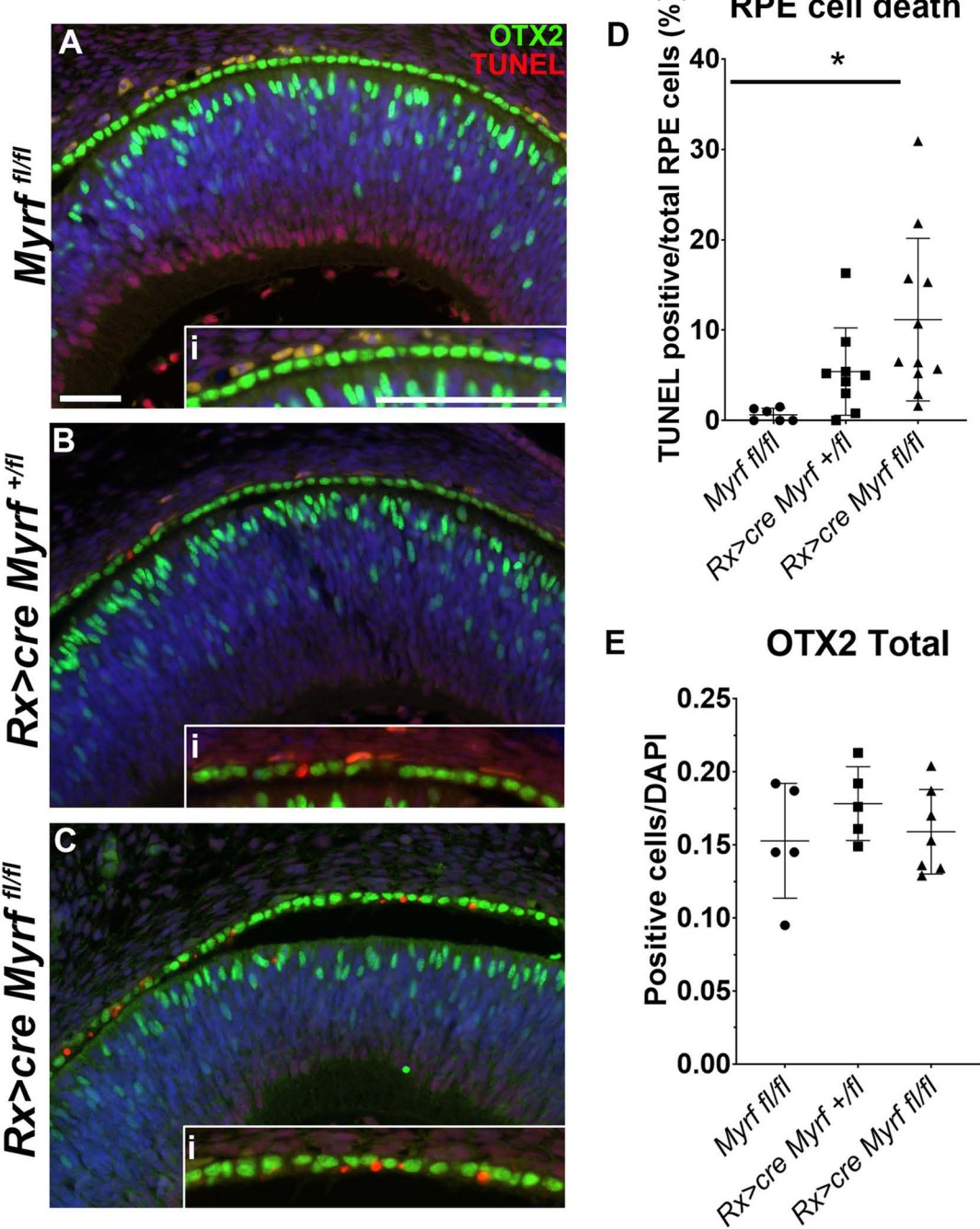

**Fig 3. RPE cell death occurs in *Rx>cre Myrf*<sup>fl/fl</sup> mutants at e14.5.** (A-C) Apoptotic cells were identified by TUNEL staining (red) in *Myrf*<sup>fl/fl</sup>, n = 7 (A), *Rx>cre Myrf*<sup>+/fl</sup>, n = 9 (B), and *Rx>cre Myrf*<sup>fl/fl</sup>, n = 10 (C) mice at embryonic day e14.5. OTX2 expression (green) was used to mark the RPE and retinal progenitor cells. Higher magnification insets were used to demonstrate the increase in TUNEL positive cells in the *Rx>cre Myrf*<sup>fl/fl</sup> RPE compared to controls (Ai-Ci). (D) TUNEL positive cells within the RPE monolayer (OTX2 positive) were quantitated and compared to the total number of DAPI+ cells within the RPE. (E) OTX2 positive cells within the RPC layer were quantitated and compared to the total number of DAPI+ cells within the RPC. One way ANOVA followed by Tukey's test was used for statistical analysis, * p < 0.05 Scale bar, 50 um.

the P0 timepoint in the *Rx>cre Myrf*<sup>fl/fl</sup> RPE (S7 Fig). The number of OTX2 expressing cells in the retina was unchanged between genotypes (Fig 3D). These studies confirm that the RPE cell population changes in *Rx>cre Myrf*<sup>fl/fl</sup> eyes are due to apoptosis, and that previously reported photoreceptor loss occurs after P0 [28].

## *Myrf* loss alters conserved biological pathways in RPE development

To define genes regulated by *Myrf* and key cellular pathways responsible for the RPE development and maturation, we conducted differentially expressed genes analysis (DEGs) within the RPE cluster at each developmental timepoint. Loss of *Myrf* is observed at all timepoints and reduction of *Tmem98* can be seen starting at e15.5, consistent with our prior report [28]. We identified many genes known to be associated with inherited retinal disease (IRD) and age-related macular degeneration (AMD) that are downregulated in the absence of *Myrf.* These include *Unc119* and *Rlbp1* starting at e13.5 and *Cdh3, Arhgef18, Ctsd, Rgr, Rdh5, Timp3, Slc16a8, Col8a1,* and *Cfh* at P0, suggesting that *Myrf* plays a key role in regulating RPE disease associated genes (Fig 4A). To further explore pathways impacted by loss of *Myrf* and define novel gene candidates for RPE disease, we analyzed enrichment of Gene Ontology (GO terms) on differentially expressed transcripts from the RPE cluster using the PANTHER Overrepresentation Test (Release 20200728) and the GO Ontology database https://doi.org/10.5281/zenodo.3954044Released2020-07-16. GO terms significantly enriched in genes that are downregulated in the *Rx>cre Myrf*<sup>fl/fl</sup> mutant include regulation of melanin biosynthetic process (MelBio, 75 fold, $p=0.005$), visual perception (VisPer, 11 fold, $p=0.05$), extracellular matrix organization (ECM, 8 fold, $p=0.02$), organic hydroxy compound metabolic process (HydCompMet, 6 fold, $p=0.01$), blood vessel development (BV, 5 fold, $p=0.02$), tube morphogenesis (TM, 5 fold, $p=0.02$), regulation of anatomical structure morphogenesis (AnaStrMor, 4 fold, $p=0.02$), and positive regulation of cellular process (RegCellPro, 2 fold, $p=0.03$) (Fig 4B). GO terms enriched in genes that are upregulated in the *Rx>Cre Myrf*<sup>fl/fl</sup> mutant include positive regulation of cellular proliferation (PosRegCellProlif, 8 fold, $p=0.02$) and cellular response to chemical stimulus (CellRespChemStim, 6 fold, $p=0.01$) (Fig 4C). Within these GO terms, underlying genes that were differentially expressed across all three timepoints were defined to compile candidates that are more likely to be direct targets of *Myrf* (Fig 4D) from the complete list of DEGs (S1 Table). Critical regulators of melanogenesis (*Pmel, Slc24a5, Tyrp1, Cdh2),* visual perception(*Lum, Cfh, Rdh10, Rdh5),* extracellular matrix (*Vit, Fbln2, Aplp1, Ramp2),* hydroxy compound metabolism *(Ttr, Rbp1, Dct),* blood vessels *(Bmpt, Thbs1, Slc1l1, Fzd4),* tube morphogenesis *(Sox10, Mgp, Lrp1),* anatomical structure *(Ezr, F3, Kif1a, Ermn),* and regulation of cellular process (*Acsk1, Cst3, Otx2, Trf*) are downregulated. Key regulators of cellular proliferation and cellular respiration (*Enpp2, Gas6, Ptn, Ctgf, Id1, Id3, Fgf15, Igfbp2*) are upregulated.

## MYRF regulates key cytoskeletal proteins and alters RPE morphology

Enrichment of melanin biosynthetic process and extracellular matrix organization suggests *Rx>cre Myrf*<sup>fl/fl</sup> mutants may have structural changes in the RPE and are consistent with the observed loss of pigmentation in *Rx>cre Myrf*<sup>fl/fl</sup> mice. In principle, loss of pigmentation could be related either to loss of melanosomes or reduced melanin. To define ultrastructural features and the source of depigmentation in *Myrf* loss-of-function, we conducted Transmission Electron Microscopy (TEM) in *Myrf*<sup>fl/fl</sup> and *Rx>cre Myrf*<sup>fl/fl</sup> mice at P21 (Fig 5). *Rx>cre Myrf*<sup>fl/fl</sup> mutants exhibit abnormal apical microvilli structure and organization, loss of photoreceptors, abnormal outer segment (OS) structure, and thickening of Bruch's membrane (Fig 5A and 5B). Additionally, *Rx>cre Myrf*<sup>fl/fl</sup> mutants displayed a significant reduction in the number of melanosomes in the RPE ($4.9\pm2.3\times10^5$/mm$^2$ vs. $1.6\pm0.7\times10^5$/mm$^2$, $p=0.0103$), suggesting loss of melanosomes contributes to the depigmentation phenotype in *Myrf* loss-of-function mice (Fig 5A-5C). Consistent with changes in ultrastructural organization, *Rx>cre Myrf*<sup>fl/fl</sup> mutants had reduced expression of the apically localized cytoskeletal protein *Ermn* [33] and extracellular matrix marker *Upk3b* (Fig 6). FeaturePlots from the scRNAseq data reveal specific expression of *Ermn* and *Upk3b* within the

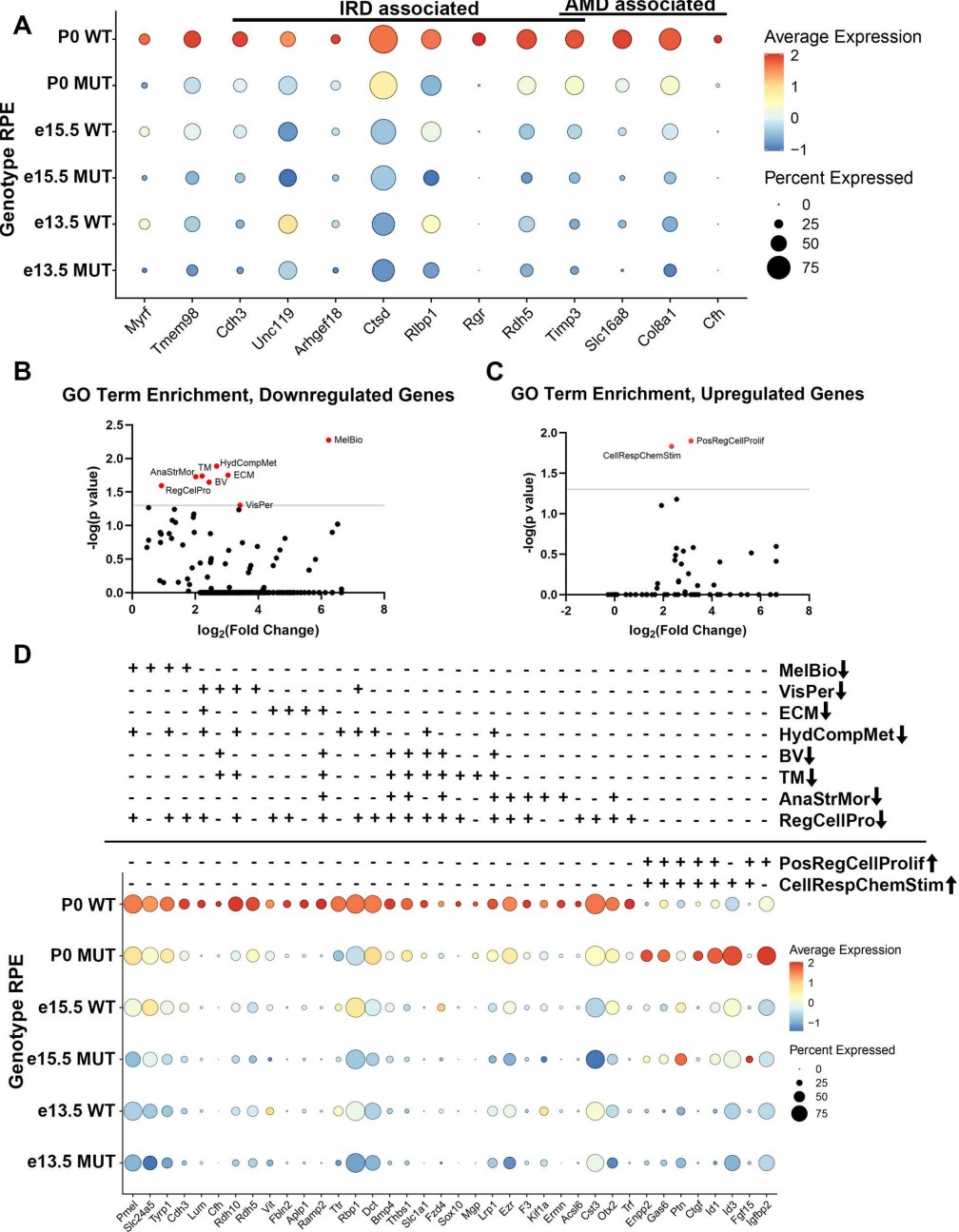

**Fig 4. Loss of *Myrf* during development alters critical pathways in RPE development and disease.** (A) DotPlot analysis showing the impact of loss of *Myrf* on expression of genes known to play a role in Inherited Retinal Diseases (IRD) and Age-related Macular Degeneration (AMD). (B-C) PANTHER GO terms enriched in downregulated (B) or upregulated (C) genes in the RPE cluster differential expression analysis of *Rx>Cre Myrf^{fl/fl}* vs. *Myrf^{fl/fl}* controls across all timepoints. (D) DotPlot analysis of selected genes from enriched GO terms across biological pathways, highlighting changes in genes involved in regulation of melanin biosynthetic process (MelBio), visual perception (VisPer), extracellular matrix organization (ECM), organic hydroxy compound metabolic process (HydCompMet), blood vessel development (BV), tube morphogenesis (TM), regulation of anatomical structure morphogenesis (AnaStrMor), positive regulation of cellular proliferation (PosRegCellProlif), cellular response to chemical stimulus (CellRespChemStim).

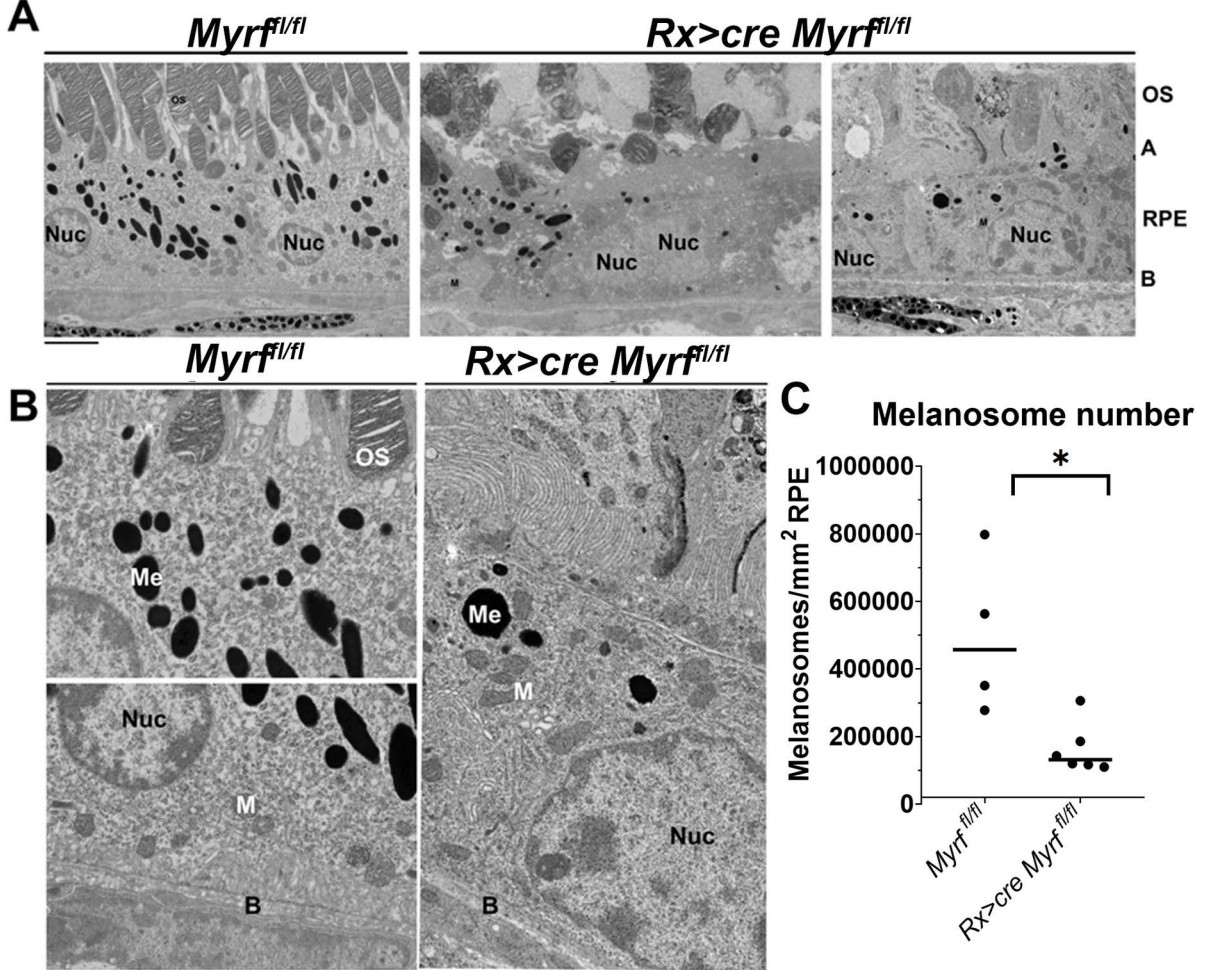

**Fig 5. Changes in ultrastructural features in *Rx>cre Myrf^fl/fl* RPE identified in TEM.** Low (A) and high (B) magnification transmission electron micrograph (TEM) images were generated to analyze the ultrastructure, and number of melanosomes (Me), Bruch's membrane (B), apical microvillous structure in the RPE and outer segment (OS) structure and in *Myrf^fl/fl* (n = 4) controls and *Rx>cre Myrf^fl/fl* (n = 6) mutants at P21. C. Quantification of the number of melanosomes per mm² RPE. Scale bar in A, 4um; in B, 1um. A, apical surface; M, mitochondria; *, $p < 0.05$.

RPE cluster in *Myrf^fl/fl* dataset and loss of expression in the *Rx>cre Myrf^fl/fl* dataset (Fig 6A), with onset of *Ermn* expression at e13.5 and increasing expression through development and onset of *Upk3b* expression at P0. To validate dysregulation of ERMN expression in *Rx>cre Myrf^fl/fl* eyes, immunohistochemical analysis was performed at P21. Co-localization of apical cytoskeletal marker ezrin (EZR) and ERMN were observed in *Myrf^fl/fl* control RPE (Fig 6B), but ERMN expression was reduced in in *Rx>cre Myrf^fl/fl* mutants, while EZR retained apical localization (4.4 fold decrease in *Rx>cre Myrf^fl/fl* mutants compared to *Myrf^fl/fl* controls, $p = 0.0013$) (Figs 6C and S8A, and S1 Data). Specificity of expression of extracellular matrix gene, *Upk3b*, to the RPE and its loss in *Rx>cre Myrf^fl/fl* eyes at P0 were confirmed by RNA scope *in situ* hybridization in the absence of a validated antibody (2.5 fold decrease in *Rx>cre Myrf^fl/fl* mutants compared to *Myrf^fl/fl* controls, $p = 0.0362$) (Figs 6D and S8B). The structural changes observed in the *Rx>cre Myrf^fl/fl* mutants and the loss of ERMN and *Upk3b* expression support a role for MYRF in RPE structure and microvilli organization.

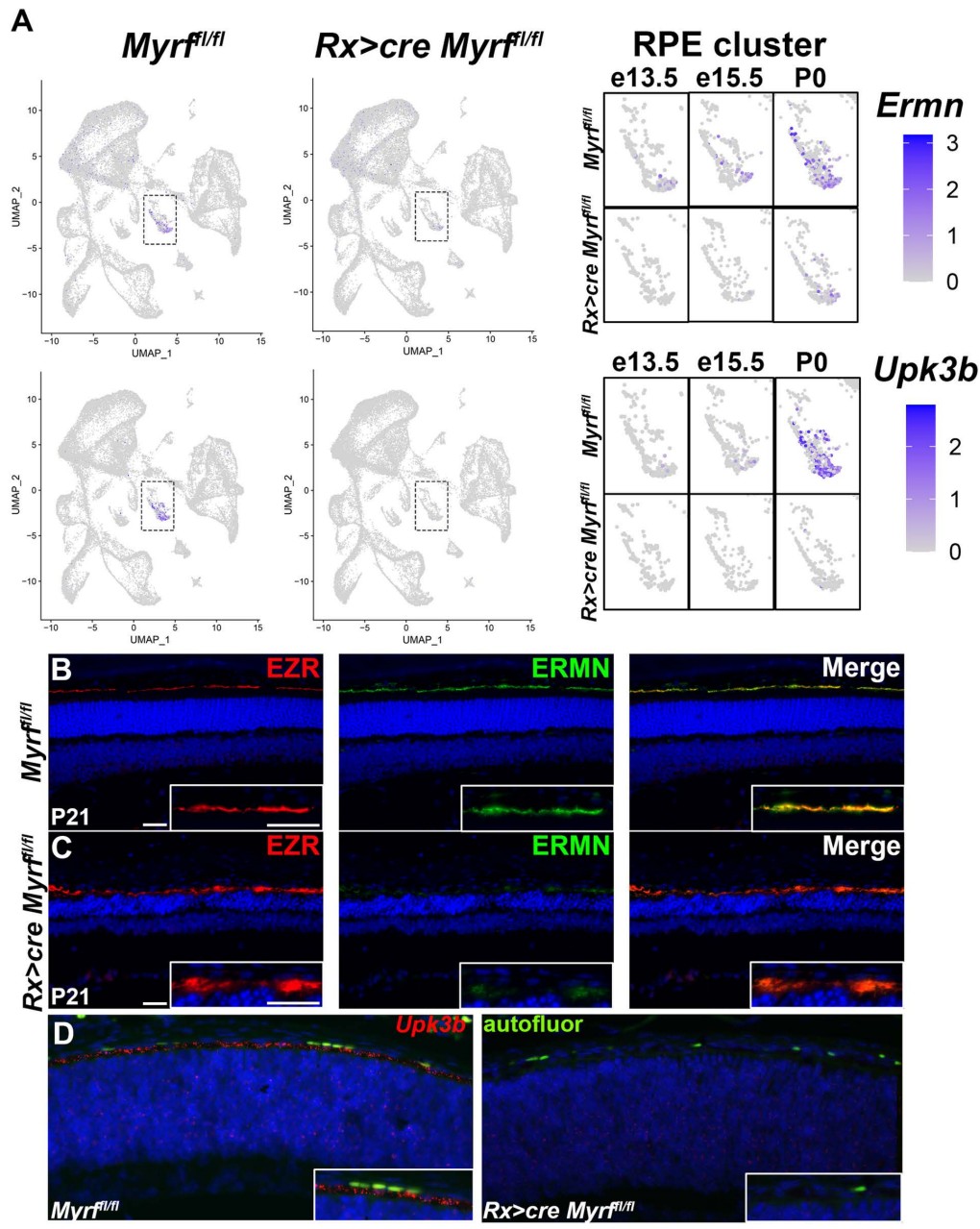

**Fig 6. MYRF regulates expression of cytoskeletal and extracellular matrix proteins.** (A) FeaturePlot analysis of expression of *Ermn* and *Upk3b* showing specificity of expression to the RPE cluster and loss of expression in the *Rx>cre Myrf*^fl/fl mutants. Larger images show expression within the RPE cluster through development. (B, C) Cytoskeletal proteins were marked within the RPE using antibodies to Ezrin (EZR, red) and Ermin (ERMN, green). (B) P21 sections of *Myrf*^fl/fl control eyes (n=2) show both EZR and ERMN localized to the RPE. (C) P21 sections of *Rx>cre Myrf*^fl/fl mutants (n=3) show retention of EZR expression and loss of ERMN expression within the RPE. (D) RNAscope of *Upk3b* (red) at P0 shows expression specific to the RPE layer which sits below the autofluorescing choroidal red blood cells (green) and loss of expression in the *Rx>cre Myrf*^fl/fl mutants (n=3) compared to *Myrf*^fl/fl controls (n=4). Insets show a magnification of the RPE layer. Scale bars in all panels represent 25 um.

## Loss of MYRF leads to altered transcription factor regulation including loss of SOX10 expression in RPE

To define the place of MYRF in the regulatory hierarchy of RPE development, we conducted Single-Cell rEgulatory Network Inference and Clustering (SCENIC) analysis [34,35]. SCENIC is a computational tool that analyzes the co-expression of transcription factors with other genes in the dataset and considering putative cis-regulatory binding motifs present in the proposed target genes to define regulons. A higher regulon activity is assigned to transcription factors co-expressed with targets containing putative binding motifs upstream of the transcription start site (Fig 7A and S2 Table). Within the RPE cluster, transcription factors known to be important in early stages of RPE development including *Vax2* and *Pax6* have high regulon activity at e13.5 and e15.5 and those important later in development including *Otx2* and *Vsx2* have higher regulon activity at P0. *Pax6, Mitf, Gsx2, Vsx2,* and *Otx2* regulons were among those unaffected by *Myrf* loss through multiple RPE developmental timepoints, suggesting that *Myrf* does not directly regulate these genes or their targets. Consistent with this, we see no significant differences in transcription factor expression levels among these genes (S9 Fig). In contrast, we see a significant reduction in *Sox10* regulon activity within the RPE at P0. The genes predicted in SCENIC to be regulated by SOX10 include *Ermn, Tmem98,* and *Wfikkn2* (S3 Table). Given that SOX10 and MYRF have been identified as co-regulators of oligodendrocyte differentiation and maturation [36,37], we next investigated whether a similar regulatory network may exist in RPE. *Sox10* transcripts were found to be significantly reduced in the *Rx>cre Myrf*^fl/fl^ mutant RPE DEG list and identified in two categories in the GO term enrichment analysis. We analyzed the distribution of *Sox10* transcripts across our scRNAseq clusters and found they were downregulated specifically in the *Rx>cre Myrf*^fl/fl^ RPE cluster compared to *Myrf*^fl/fl^ controls across all time points but not changed in the melanocyte cluster (Fig 7B). Consistent with these results, SOX10 protein is expressed in both the melanocytes and RPE of *Myrf*^fl/fl^ controls at e13.5, e15.5, and P3 (Fig 7C-7E). SOX10 positive cells in the RPE were quantitated and compared to the total number of DAPI cells (S10 Fig). One-way ANOVA followed by Tukey multiple comparisons was used to show a statistically significant decrease in SOX10 positive cells in *Rx>cre Myrf*^fl/fl^ RPE at all timepoints analyzed (S1 Data). While SOX10 expression is retained in the melanocytes of *Rx>cre Myrf*^fl/fl^ mutants, it is absent from the RPE cells. We have confirmed the specificity of *Sox10* expression in our scRNAseq dataset using Pseudobulk RNAseq analysis of the RPE and melanocyte clusters (S4 Table). These results suggest that *Sox10* expression is dependent on *Myrf* within the RPE.

## MYRF regulates genes involved in proliferation, including BMP/TGFß signaling

Further investigation of regulon dysregulation in the RPE revealed many members of the BMP/TGFß signaling pathway, including *Smad5, Tgif2, Smad4,* and *Smad1* showing differential regulon activity between the *Myrf*^fl/fl^ and *Rx>cre Myrf*^fl/fl^ RPE clusters (Fig 7A). Given the role of BMP/TGFß signaling in cellular communication and differentiation, we used the computational tools MultiNichenetr [38] and Differential Nichenetr [39] to predict ligand, receptor, target gene interactions within the RPE cluster (Figs 8A and 8B, and S11). Consistent with the SCENIC analysis, differential interactions within the BMP/ TGFß signaling were also identified, including *Bmp4, Tgfb2,* and *Bmp2*. Interestingly, the expression and ligand activity of *Bmp2,* a known regulator of eye growth, is increased in the mutants, and multiple targets are also present within the RPE cluster. Expression analysis of BMP/TGFß signaling pathway members within the RPE cluster throughout development (Fig 9A) revealed dysregulation of TGFß/BMP components and effectors within the RPE. A known inhibitor of the pathway, *Wfikkn2,* and its downstream target, *Gdf11,* are decreased in the *Rx>cre Myrf*^fl/fl^ RPE clusters. Consistent with the loss of inhibition, increases in expression of *Tgfb2, Bmp2, Smad1, Smad6, Smad9, Id1, Id2,* and *Id3* were also observed. VlnPlot analysis of the RPE cluster at P0 highlights a significant decrease in expression of *Wfikkn2* in the mutants (p_adj = 1.36E-16) and increases in *Tgfb2* (p_adj = 5.53E-10) and *Id3* (p_adj = 1.43E-37). (Figs 9B, and S12E). We confirmed *Wfikkn2* was significantly decreased in the *Rx>cre Myrf*^fl/fl^ mice at P0 using RNAscope with quantification of immunofluorescence intensity compared to background intensity per area in ImageJ (Figs 9C, and S12A, and S1 Data) TGFB2 is expressed in the RPC layer of *Myrf*^fl/fl^ eyes at P0, with little to no expression in the RPE layer (Fig 9D). TGFB2 expression was observed in both the RPE and RPC layers of *Rx>cre Myrf*^fl/fl^ mutants, with enriched levels in the RPE

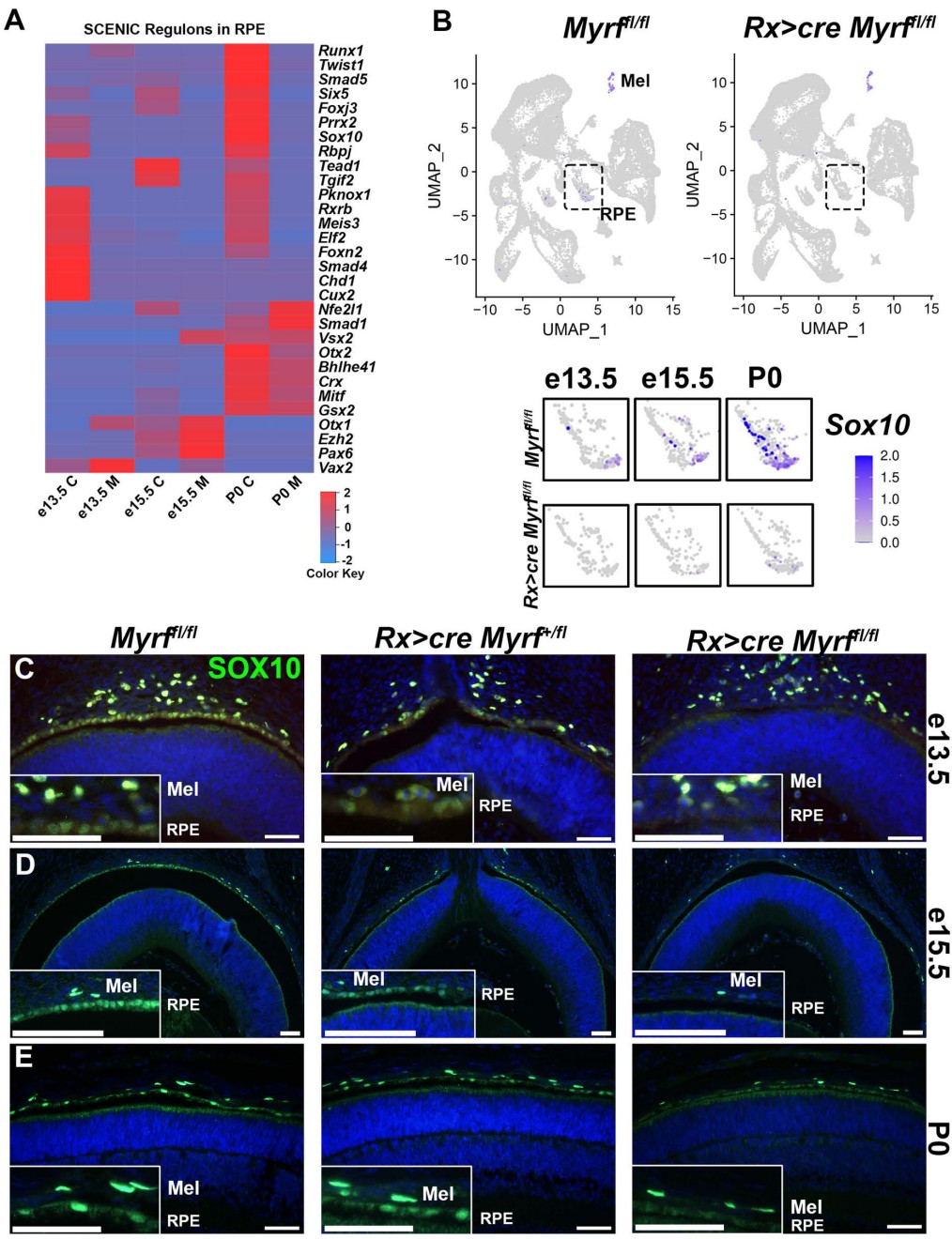

**Fig 7. *Myrf* regulates SOX10 expression in RPE cells.** (A) Single-Cell rEgulatory Network Inference and Clustering (SCENIC) analysis within the RPE cluster from the scRNAseq data infers active transcription factors across all developmental time points including members of the TGFB signaling pathway. (B) FeaturePlot analysis of scRNAseq transcripts shows *Sox10* expression in both the melanocyte and RPE clusters in *Myrf^{fl/fl}* controls. Expression is retained in the melanocyte cluster in *Rx>cre Myrf^{fl/fl}* mutants and lost in the RPE cluster. Larger images of the RPE cluster demonstrate onset of expression in the *Myrf^{fl/fl}* controls at e13.5 and increasing through P0. Only a few cells containing *Sox10* transcripts can be identified in the *Rx>cre Myrf^{fl/fl}* mutant RPE at later time points. Larger image of the melanocyte cluster shows no changes in expression. (C-E) SOX10 protein is localized to the melanocytes and RPE in *Myrf^{fl/fl}* and *Rx>cre Myrf^{+/fl}* controls at e13.5 (*Myrf^{fl/fl}* (n = 3), *Rx>cre Myrf^{+/fl}* (n = 4), *Rx>cre Myrf^{fl/fl}* (n = 4)) (C); e15.5 (*Myrf^{fl/fl}* (n = 3), *Rx>cre Myrf^{+/fl}* (n = 3), *Rx>cre Myrf^{fl/fl}* (n = 3)) (D); and P3 (*Myrf^{fl/fl}* (n = 3), *Rx>cre Myrf^{+/fl}* (n = 2), *Rx>cre Myrf^{fl/fl}* (n = 3)) (E). SOX10 expression is only retained in the melanocytes of *Rx>cre Myrf^{fl/fl}* mutants. Scale bars, 50um.

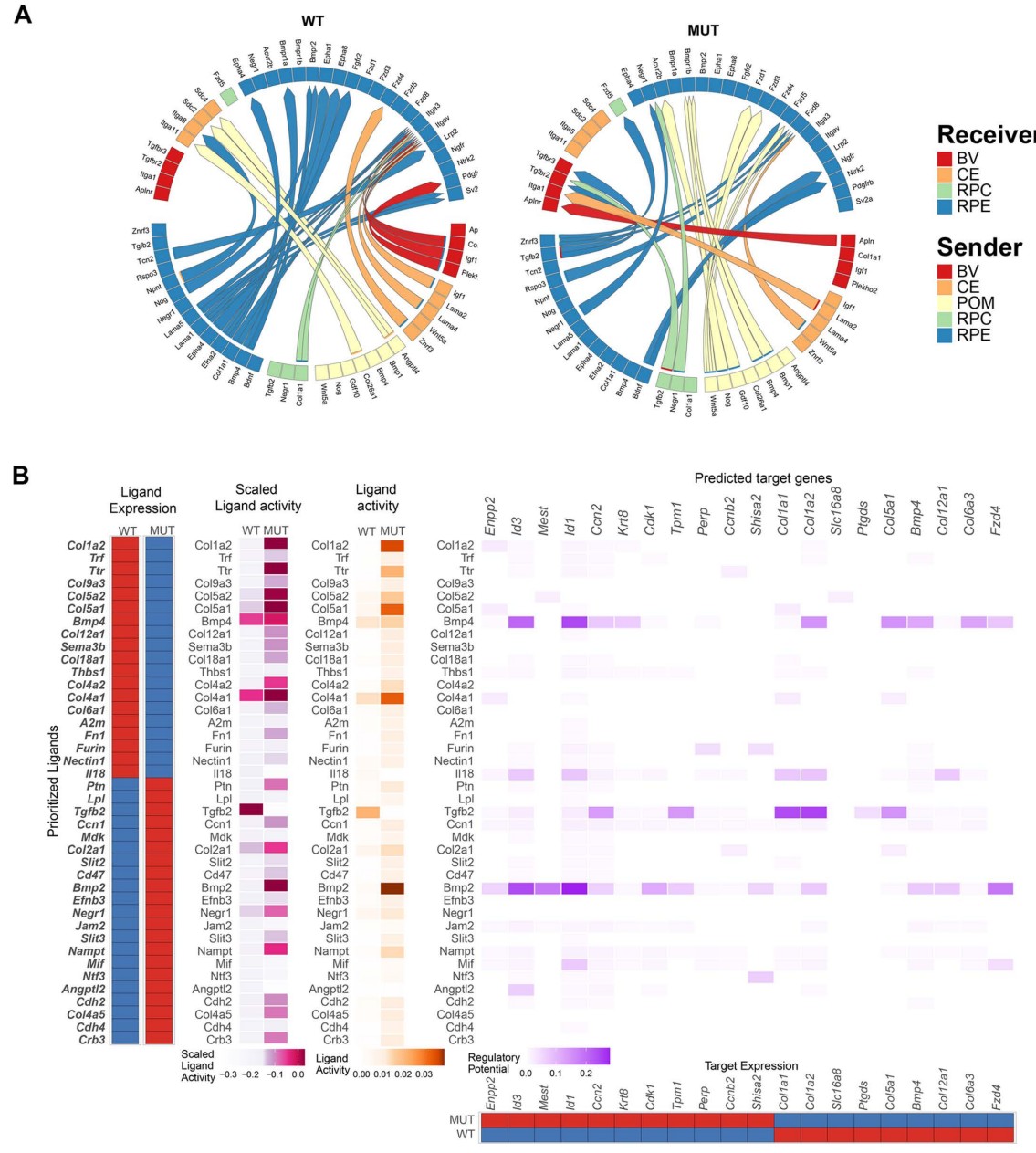

**Fig 8. Computational analysis of the RPE cluster suggests alterations in the TGFB signaling pathway.** (A) Circos plot generated from MultiNichenetr analysis predicting ligand receptor partners in control and mutant scRNAseq data sets from a combination of senders and receivers. (B) Analysis of changes in predicted signaling pathways within the RPE cluster using Differential Nichenetr. Expression and activity of ligands within the RPE is seen on the left and predicted targets within the RPE based on the Differential Nichenetr pathway analysis is seen on the right side.

(S12B Fig, and S1 Data). Elevated levels of *Bmp2* transcripts were also detected in the RPE layer using *Bmp2* RNAscope (S12F and S12G Fig, and S1 Data). Consistent with increases in BMP/TGFß signaling, the *Bmp2* inducible transcription factor *Id3*, showed increased expression within the RPE cluster of *Rx>cre Myrf*^fl/fl^ mutants (Fig 9E), and quantitation of *Id3* RNAscope confirmed a statistically significant increase (Figs 9E and S12C, and S1 Data). Activation of BMP/TGFß signaling leads to phosphorylation of activating SMAD transcription factors (i.e., SMAD1/5/9). We investigated phosphorylated

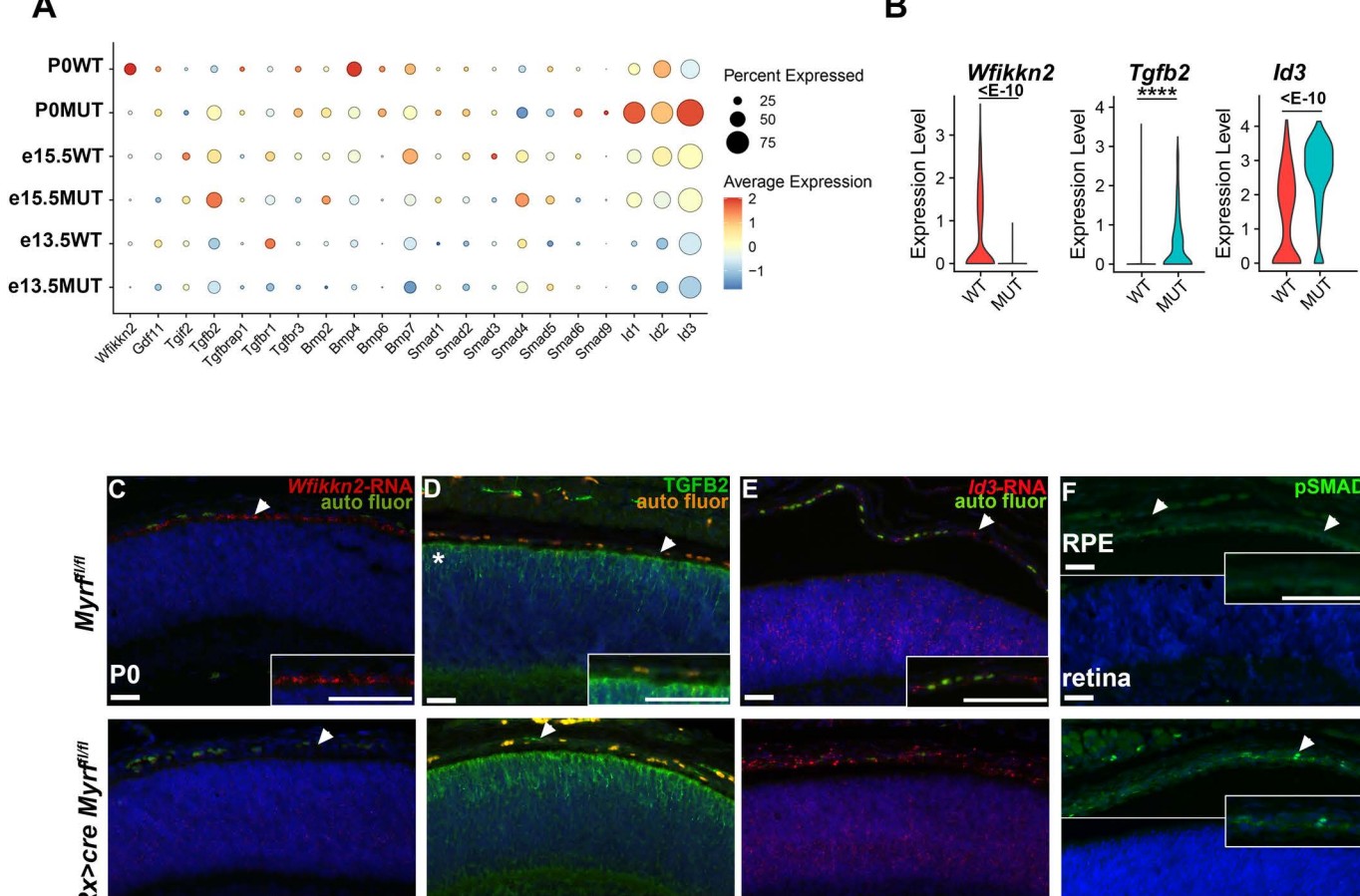

**Fig 9. Increased BMP/TGFB signaling pathway activity in *Rx>cre Myrf*<sup>fl/fl</sup> mutants.** (A) DotPlot analysis of TGFB family members expressed in the RPE cluster across all genotypes highlights differential expression between *Myrf*<sup>fl/fl</sup> and *Rx>cre Myrf*<sup>fl/fl</sup> cells. (C-*Myrf*<sup>fl/fl</sup>; M-*Rx>cre Myrf*<sup>fl/fl</sup>) (B) VlnPlot representation of transcript levels of *Wfikkn2*, *Tgfb2,* and *Id3* in theRPE cluster highlighting a decrease in the pathway inhibitor, *Wfikkn2* and increases in *Tgfb2* and *Id3*. (C) RNAscope of *Wfikkn2* confirms specificity to RPE layer in *Myrf*<sup>fl/fl</sup> (n=4) and loss of expression in the *Rx>cre Myrf*<sup>fl/fl</sup> mutants (n=3) at P0. (D) TGFB2 protein is observed in the RPC of *Myrf*<sup>fl/fl</sup> controls (n=4) at P0, but not in the RPE layer. TGFB2 is present in the RPE layer of *Rx>cre Myrf*<sup>fl/fl</sup> mutants (n=4) (E) RNAscope of *Id3* shows increased expression in the RPE layer of *Rx>cre Myrf*<sup>fl/fl</sup> mutants (n=5) compared to controls (n=3). (F) Phosphorylated SMAD 1/5/9 immunostaining indicates an increase in activity within the RPE layer of *Rx>cre Myrf*<sup>fl/fl</sup> mutants (n=5) compared to no staining in *Myrf*<sup>fl/fl</sup> controls (n=4). (RPC- retinal progenitor cells; RPE- retinal pigmented epithelial cells; arrowheads-RPE layer, asterisk-RPC layer). Insets show a magnification of the RPE. Auto fluorescing blood cells are labeled at auto fluor in the appropriate panels. Scale bars in all panels indicate 50um.

SMAD, pSMAD, as a marker of signaling activity. pSMAD 1/5/9 was detected at low levels at P0 in *Myrf*<sup>fl/fl</sup> controls (Fig 9F) and significantly increased in the RPE of *Rx>cre Myrf*<sup>fl/fl</sup> mutants (S12D Fig, and S1 Data). Together our results support that loss of *Myrf* leads to activation of BMP/TGFß signaling and its effectors, linking *Myrf* to RPE dysfunction and/or eye growth pathways.

## Discussion

The exact localization and function of *Myrf* during eye development is controversial. We have provided multiple lines of evidence supporting RPE predominant expression of *Myrf* during mouse ocular development, suggesting its primary role is within the RPE. Our initial studies with human *MYRF* expression demonstrated RNA specificity to the RPE and optic

nerve, with no transcripts identified in the retina or ciliary body by qPCR [28]. We also confirmed in mouse that *Myrf* transcripts were over 100-fold more abundant in RPE as compared to retina [28]. In addition, we provided antibody staining in mice and human samples, depicting specific staining in the RPE and high non-specific background levels throughout the retina even in *Myrf* knockout mice. Compiled aggregated RNAseq data from the National Eye Institute EyeIntegration Pan Human Gene Expression plots [40] support exponentially higher levels of *MYRF* expression in the RPE compared to the cornea or retina (S13 Fig). Our current data with RNAscope *in situ* hybridization in control and *Rx > cre Myrf*$^{fl/fl}$ mutants throughout development and postnatal time points supports that *Myrf* transcripts are specifically found in the RPE at least by E13.5 and show loss of expression in the *Myrf* conditional knockout mice. Further, our scRNAseq data detect *Myrf* transcripts only in the RPE cluster. The vast number of mouse models in which defects in the RPE lead to photoreceptor impairment (*Atg5* and *Atg7* [41], *Ift20* [42], *Mfrp* [43], *Dapl1* [44], *Tsc1* [45], *Kcnj13* [46], *Sod2* [47], *Rb1cc1* [48], *Lamp2* [49], *Bsg* [50], *Pten* [51], and *RPE65* [52]) and the preservation of RPC progenitors and early photoreceptors in our *Rx > Cre Myrf*$^{fl/fl}$ mice support that phenotypes in these mice are primarily RPE-driven. We have also shown that deletion of *Myrf* using RPE specific cre, *RPE-Tyrcre > ERT2,* recapitulates the phenotype observed with the *Rx > cre* deletion, supporting the role of MYRF in the development of the RPE.

Our results support that *Myrf* is vital for cell survival, melanogenesis, and establishing structural integrity of RPE. Other mouse models of reduced *Myrf* in eye development, including one model of a heterozygous patient variant and one heterozygous for a loss of function allele, suggest MYRF may play a role in the ciliary body and retina, respectively [29,30]. Based on the age of mice analyzed, the phenotypes observed in these mouse models may be secondary to loss of MYRF in the RPE or a result of off target CRISPR effects. The genes altered in these mouse models are not different in our scRNAseq dataset (S14 Fig). In our model, RPE specification is unaffected in *Rx > cre Myrf*$^{fl/fl}$, based on the initial formation of a pigmented monolayer, no changes in early marker expression, and no evidence of neural retina transdifferentiation. However, we observe significant gene expression changes by scRNA sequencing even at early time points. Apoptotic RPE cells are identified in the *Rx > cre Myrf*$^{fl/fl}$ mutants during embryonic development and an overall reduction in proportion of RPE cells in the eye cups is observed, preceding the loss of pigmentation. The RPE pigmentation defects appear to be driven by loss of melanosomes number and reductions in expression of melanosome associated genes, such as *Pmel* and *Tyrp1*.

Our ultrastructural and gene expression analysis reveals disruption and disorganization of the microvilli on the apical surface of the RPE cells, which leads to loss of their intimate contact with the photoreceptor outer segments. This interaction is critical to the function of the RPE cells in phagocytosis of photoreceptor outer segments and contributes to the degeneration of the retinal layer. A thickening of Bruch's membrane can also be seen in the *Rx > cre Myrf*$^{fl/fl}$ mutants. A similar increase in thickness of Bruch's membrane is also observed in *Lamp2* deficient mice as the RPE ages and is thought to be coincident with impaired phagocytosis and accumulation of lipid deposits [49]. These features are often seen in age-related macular degeneration (AMD) and RPE-driven inherited retinal diseases. Indeed, we show that conditional loss of *Myrf* impacts the expression of many genes known to be involved in IRD and AMD. We have identified novel targets of *Myrf* that impact RPE cytoskeleton and ultrastructure that may contribute to these structural phenotypes, including *Ermn* and *Upk3b. Ermn,* encodes for Ermin (ERMN), an F-actin binding protein in the ezrin, radixin, moesin superfamily. While it has not been identified as a direct target of *Myrf* in oligodendrocytes [53], it plays a critical structural role in myelination, and induces cellular protrusions in oligodendrocytes [54]. In the RPE, ERMN is specifically expressed in the RPE of adult rat and colocalizes with cytoskeletal markers in the rat and in ARPE19 cells, although its exact role is unknown [33]. We show that ERMN expression is likewise localized to the apical surface of the RPE, and expression is lost in *Myrf* knockouts. Heterozygous *ERMN* variants have recently been identified in a family with multiple sclerosis, but ocular features such as retinal degeneration or nanophthalmos were not ascertained in this family [55]. *Upk3b* has not been studied in eye development but is a component of the urothelium, which is an epithelial structure composed of extracellular matrix and provides and extensive barrier protecting the urethra from the toxicity of urine [56]. It is intriguing to consider its importance in the RPEas the RPE is critical to forming the blood-retinal barrier.

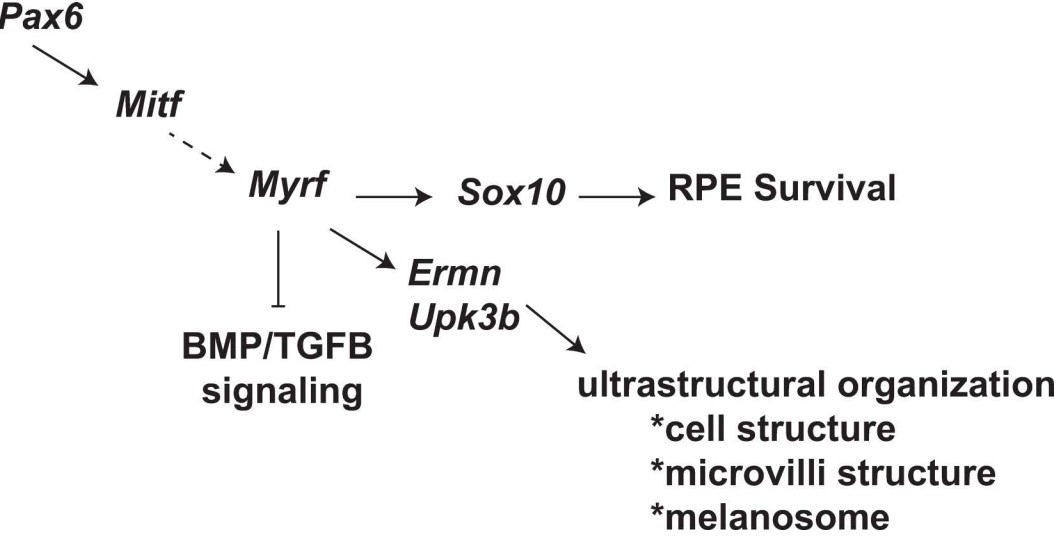

**Fig 10. Model.** Transcriptional and phenotypic analysis of the *Rx>cre Myrf*<sup>fl/fl</sup> mice suggests *Myrf* is downstream of *Pax6* and *Mitf* in the hierarchy of RPE transcription factor regulation. *Myrf* is important for the activation of *Sox10* which may impact RPE cell survival, *Ermn* and *Upk3b*, which may regulate ultrastructural organization of the RPE, and BMP/TGFB signaling, which may influence eye growth or RPE dysfunction.

Based on the expression of *Myrf* in the RPE and the role it plays in RPE development, placing it in the hierarchy of transcription factors is important for understanding its impact in RPE development and disease, and potential as a therapeutic target for RPE degeneration and eye size disorders. The temporal hierarchy of expression of many of the transcription factors known to be involved in RPE development has been established (reviewed in [3,57,58]). *Pax6* activates *Mitf* and together they are important for early activation of genes involved in melanogenesis, as well as specification of the RPE cell fate [59]. Based on our data we propose a model placing MYRF downstream or parallel to PAX6 and MITF (Fig 10). This is supported by 4 observations. First, expression of *Pax6* and *Mitf* are not changed in our scRNAseq dataset between *Rx>cre Myrf*<sup>fl/fl</sup> and controls. Second, analysis of SCENIC regulon activity in the RPE cluster of our scRNAseq data, predicts comparable regulon activity of the early RPE transcription factors *Pax6, Mitf,* and *Otx2* in wild-type and *Rx>cre Myrf*<sup>fl/fl</sup>. Third, global loss of PAX6 or MITF in mice leads to microphthalmia or ocular disorganization and disruption of the early eye field [60–62], while early loss of *Myrf* in our animal model still leads to normal eye morphogenesis. In our study, we show that RPE cells are specified in *Rx>cre Myrf*<sup>fl/fl</sup> eyes even though they have similar melanogenesis defects as *Pax6* and *Mitf* knockout mice. Fourth, *Tead1*, a downstream target of the Hippo pathway, was also identified as a regulon active during early RPE development and unchanged during early development in our *Rx>cre Myrf*<sup>fl/fl</sup> mice. Conditional deletion of YAP in the RPE of mice using *Rx>cre* results in transdifferentiation of the RPE to retinal-like cells, and similar transcriptional changes and loss of melanogenesis are observed in yap -/- zebrafish as in our *Rx>cre Myrf*<sup>fl/fl</sup> mice [16,63]. Together, these results suggest that *Myrf* is also downstream of the Hippo signaling pathway, as similar transcriptional and phenotypic changes are observed but not transdifferentiation. An intriguing finding of our study is that expression of *Sox10* is absent specifically in the RPE of *Rx>cre Myrf*<sup>fl/fl</sup> mutants. *Myrf* and *Sox10* have been shown to be interacting partners in oligodendrocyte differentiation [64,65]. *Myrf* has been found to be a direct target of *Sox10* and they work together to switch target genes of oligodendrocyte progenitors to that of differentiation [64]. In the eye, Sox10 expression is specifically lost in RPE in our *Rx>cre Myrf*<sup>fl/fl</sup>, while melanocyte expression remains (Fig 7). Future studies with CUT&RUN or other chromatin immunoprecipitation techniques will help to determine whether these are direct targets and similar regulatory programs act in RPE development as in oligodendrocyte maturation.

We have identified an elevation of TGFβ and BMP signaling in our conditional loss of *Myrf* model, with upregulation of *Tgfb2, Bmp2* expression and downstream signaling. These pathways are particularly relevant to retinal and RPE development, RPE function, and eye size. In early stages, TGFβ/BMP signaling from the surface ectoderm has been shown to drive neural retina specification [3]. WNT and TGFβ/BMP signaling from the extraocular mesenchyme and surface ectoderm drive RPE cell fate [8]. In the chick, after removal of the surface ectoderm, application of WNT or BMP-soaked beads is sufficient to activate *Mitf* and initiate RPE specification [8]. Studies in chick and mice have also shown that BMP2 is a negative regulator of ocular growth [13,15]. Upregulation of the TGFβ pathway, specifically *Tgfb2*, has been associated with RPE dysfunction (reviewed in [66]). In RPE disease state, TGFβ signaling upregulation induces an inflammatory response, promoting an epithelial to mesenchymal transition (EMT) through both a SMAD dependent and independent pathway [67]. Tgfb2 induces EMT in cultured human ARPE19 cells [68]. In our model, condition loss of *Myrf* leads to disruption of cell structure and microvilli structure in the RPE. We speculate that the activation of the TGFβ family in *Rx > cre Myrf$^{fl/fl}$* mutants is a response to the induced disease state of the RPE.

Collectively, our study defines a critical role for MYRF in RPE development in melanogenesis, cell structure, and cell survival, places MYRF in the hierarchy of RPE differentiation, and identifies novel candidate genes for RPE-driven disorders. Future studies will be essential to parse out direct targets of *Myrf*, its role in maintaining RPE structure, and the disparate nature of the mouse and human phenotypes.

## Materials and methods

### Ethics statement

Experiments with mice were approved by the Institutional Animal Care and Use Committee at the University of Michigan.

### Mice

*Rx > cre* [69] *Myrf flox* [70], *EIIAcre* (Jackson Laboratory, JAX:003724), and *RPE-Tyrcre>ERT2* [71] mice were housed at the University of Michigan in accordance with guidelines from the Unit for Laboratory Animal Medicine and the Institutional Animal Care & Use Committee. Mice were genotyped from DNA isolated from tail biopsies, as previously described [28,71]. *EIIAcre* transgenics were bred to the *Myrf$^{fl/fl}$* mice to generate a null allele and increase the efficiency of excision when bred to the *RPE-Tyrcre>ERT2* strain. Mice were genotyped for the cre allele using the primers forward 5'GCATAACCACTGAAACAGCATTGCTG3' and reverse 5'GGACATGTTCAGGCATCGCAAGGCG3'. A 350 bp PCR product was amplified with the following conditions: 94°C 3 minutes, followed by 32 cycles of 94°C 30 seconds, 55°C 30 seconds, 70°C 30 seconds, and a final extension at 70°C for 10 minutes. The *Myrf* null allele was amplified using the primers forward 5'GCTCTTGAGTGGGGAAGCAT3' and reverse 5'GTGTATGCCAAGCTGAGGGT3'. The flox allele (1450 bp), wild-type allele (1382 bp), and the null allele (770 bp) were amplified with the following conditions: 92C 3 minutes, followed by 30 cycles of 92°C 10 seconds, 57°C 30 seconds, 72°C 30 seconds, and a final extension at 72°C for 30 seconds.

### RNAscope *in situ hybridization*

Heads were harvested from embryonic day 14.5 (e14.5) embryos, dissected eyes were harvested from postnatal day 0 (P0) – P21 pups, and optic nerves were harvested from P21 pups. For larger eyes (after P0), a hole was punctured through the cornea to allow penetration of the fixative. Samples were fixed in 4% buffered paraformaldehyde (0.1M NaPO$_4$ pH7.3) for 0.5-4 hours at 22°C, dehydrated through increasing concentrations of ethanol up to 70%. The samples were then embedded in paraffin using the TissueTek VIP Model VIP5A-B1 (Sakura Finetek USA, Inc.) and the Shandon Histocentre 2 Model #64000012 embedding station (Thermo Fisher Scientific) and sectioned at 5–6μm.

For *in situ* hybridization, RNAscope was performed using the RNAscope Multiplex Fluorescent Detect V2 system (Advanced Cell Diagnostics [ACD], #323110). Briefly, paraffin was removed with two changes of Xylene and then washed in 100% ethanol (ETOH).

Sections were treated with the hydrogen peroxide reagent for 10 minutes followed by two washes in distilled water. Target retrieval was performed using boiling 1X Target Retrieval Reagent for 7 minutes, followed by washing in distilled water and 100% ETOH. After drying the slides, the sections were treated with Protease Plus in prewarmed humidity chamber for 25 minutes at 40°C then washed in distilled water. Prewarmed RNAscope probes were then applied, and slides were incubated in a humidity chamber at 40°C for 2 hours. The probes used were *Mm-Myrf* (ACD, 524061), *Mm-Upk3b* (ACD, 568561), *Mm-Wfikkn2* (ACD, 531321), *Mm-Id3* (ACD, 445881), *Mm-Bmp2-E3* (ACD, 427341), and a negative control probe (ACD, 320871). After hybridization of the probe, sections were washed in 1X Wash Buffer then incubated at 40°C with Amp1 reagent for 30 minutes, Amp2 reagent for 30 minutes, and Amp3 reagent for 15 minutes, with washes in 1X Wash Buffer between each step. For signal development, the sections were then incubated in HRP-C1 for 15 minutes at 40°C, washed in 1X Wash Buffer and then incubated with Cyanine 3, Opal-Tm 570 (Akoya Biosciences, FP1488001KT) fluorophore diluted 1:1500 in TSA plus buffer (ACD, 322809) for 30 minutes at 40°C. Sections were then washed in 1 X Wash Buffer then treated with HRP Blocker for 15 minutes at 40°C, stained with DAPI (Sigma, MBD0015) and mounted in ProLong Gold Antifade (Invitrogen, P36930). Images were quantitated with ImageJ, measuring the integrated density of signal in the RPE compared to background staining per area. P value was calculated using the unpaired T test.

## Single cell RNA sequencing

Eyes were collected from e13.5, e15.5, and P0 litters, the cornea, lens, and optic nerve were removed, and the eyecups were place on ice in 1 X HBSS (Invitrogen, 14175095) while samples were genotyped using a rapid genotyping protocol [72]. Three *Myrf*<sup>fl/fl</sup> and three *Rx>cre Myrf*<sup>fl/fl</sup> samples were pooled for each time point. Eyecups were dissociated into single cells in a Papain solution containing 5mM L Cysteine (Sigma, C7352), 1mM EDTA (Sigma, E4884), 0.6mM 2-mercaptoethanol (Sigma, M6250), and 1mg/ml Papain (Roche, 10108014001) for 10 minutes at 37°C with trituration every 2 minutes. The dissociation was stopped using Neurobasal Media (Invitrogen, 12348017) with 10% Fetal Bovine Serum (FBS) (Corning, MT35010CV), single cells were collected by centrifugation for 5 minutes at 300 RCF at 4°C. The supernatant was aspirated, and the single cells were resuspended in cold Neurobasal Media with 3% FBS. Single cell libraries were prepared by the University of Michigan Advanced Genomics Core and sequencing was performed using the 10X Genomics Chromium platform using manufacturer's protocol for 10x Single Cell Expression 3' and sequenced on the NovaSeq (S4) 300 cycle (Illumina, San Diego, CA). Sequencing outputs were demultiplexed using 10x Genomics Cell Ranger 7.1.0 software and FASTQ files were aligned to the Genome Reference Consortium Mouse Build 38, mm10 genome [73–75]. Seurat/4.1.1 [76,77] scRNAseq software was utilized to normalize data using the NormalizeData and FindVariableFeatures commands with nfeatures set to 2000. Poor quality cells were removed using cutoffs of nFeature_RNA<200 and % mitochondrial RNA>15%. Doublet cells were considered those with nCounts>2000 and removed from the study. Data from all the time points and genotypes was anchored and integrated prior to scaling, principal component analysis and Uniform Manifold Approximation and Projection (UMAP) clustering. The FindAllMarkers command was used to identify and define unique transcripts from each cluster, and cluster identities were defined and assigned using published literature. The FindMarkers command was used to identify differentially expressed genes between clusters and genotypes. Data was displayed in RStudio using the VlnPlot, DotPlot, and FeaturePlot (reduction="umap") functions. Gene set enrichment analysis was performed using Gene Ontology Consortium website (https://www.geneontology.org/) using the PANTHER Overrepresentation Test (Release 20200728) and the database GO Ontology database https://doi.org/10.5281/zenodo.3954044Released2020-07-16 [78,79].

## Bioinformatics tools

Molecular pathway analysis was performed with Differential Nichenetr which analyzes ligand-receptor networks and generates ligand-target matrix [39,80], using the RPE cluster as the sender and the RPE cluster as the receiver. Analysis of regulon activity in the RPE cluster, the expression of a given transcription factor compared to the expression of target genes with binding sites, was performed with Single Cell rEgulatory Network Inference and Clustering (SCENIC), using

default parameters described in original manuscript [34,81]. MultiNichenetr and Differential Nichenetr were performed with R packages from the Saeys Lab, using codes provided [38,39,82]. Pseudotime analysis was performed with the Monocle3 package, using codes provided [83,84].

## Immunohistochemistry and OTX2/TUNEL staining

Antibody staining was performed using established methods [28], with the antibody conditions can be found in S5 Table. All antibodies were incubated at 4°C overnight. Nuclei for all samples were stained with DAPI (Sigma, MBD0015) and mounted in ProLong Gold Antifade Mountant (Invitrogen, P36930). ERMN immunostaining was quantitated using ImageJ. The integrated density of ERMN signal was normalized to the integrated density of EZR signal. The fold change was calculated between *Rx>cre Myrf*$^{fl/fl}$ mutants and *Myrf*$^{fl/fl}$ controls. The unpaired T Test was used to calculate the p value. Co-staining of the rabbit anti-OTX2 (Abcam, ab21990) and apoptotic cells using In Situ Cell Death Detection Kit, TMR red (Roche, 12156792910) was performed on e14.5 paraffin sections. OTX2 immunohistochemistry was performed as described above. Apoptotic cells were identified using the protocol from the In Situ Cell Death Detection Kit, TMR red. Sections pretreated with DNaseI were used as a positive control and sections without the Enzyme Solution were used as a negative control. Nuclei were stained with DAPI (Sigma, MBD0015) and mounted in ProLong Gold Antifade Mountant (Invitrogen, P36930). OTX2 positive, TUNEL positive, and DAPI positive cells were counted using Fiji, ImageJ software. A minimum of two slides were counted per sample. For the OTX2 total counts, the data is displayed as the total number of the OTX2 stained cells per total number of DAPI cells in the region counted. For the RPE apoptosis counts, the data is displayed as the total number of TUNEL positive cells per the total number of RPE cells, marked by OTX2. Each point on the graph represents an individual sample. Statistical analysis was performed using Ordinary one-way ANOVA with Tukey multicomparisons.

## Electron microscopy

Eyes were collected from P21 mice and fixed in 2% paraformaldehyde 2% glutaraldehyde in 100 mM cacodylate buffer overnight. The tissue was then rinsed in PBS, post-fixed in osmium tetroxide for 1 hour. Tissues with rinsed 3X in PBS followed by graded alcohol dehydration, and 2X rinses in propylene oxide for 10 minutes each. Eyecups were infiltrated with Epon embedding media by incubation in with propylene oxide:Epon mixes 3:1 then 1:1 then 3:1 each for 1 hour and subsequently 100% Epon overnight while gently agitating the tissue. The eyecups were then embedded in a beam capsule with 100% Epon and incubated at 60°C overnight to cure the resin. The eyecups were then cut into semithin (500 nm) section using a Ultracut E ultramicrotome (Leica Biosystems). Sections were stained with toluidine blue to identify key target areas. The block was then trimmed into 1 mm section for further EM processing. Ultrathin sections (70–90 Å) were collected with a diATOME diamond knife, placed on cooper grids, and counterstained with Uranyless (samarium/gadolinium triacetate) and lead citrate and imaged on the JEOL JSM 1400 Plus transmission electron microscope (JEOL, Peabody, MA) at the Microscopy and Image Analysis Laboratory. Melanosomes were identified within the high-resolution image by inspection and cells were manually segmented to define the cell area. The number of melanosomes per mm² were calculated. Comparison among genotypes was done by student's T-test.

## Supporting information

**S1 Fig. Deletion of *Myrf* in the RPE with the *RPE-Tyrcre>ERT2* strain recapitulates phenotype of *Rx>cre Myrf*$^{fl/fl}$ model.** *RPE-Tyrcre-ERT2>Myrf*$^{fl/-}$ and controls were injected with tamoxifen at e11.5, e12.5, and e13.5, and harvested at e18.5. (A) Pigmentation was reduced in the RPE of *RPE-Tyrcre-ERT2>Myrf*$^{fl/-}$ mice compared to *Myrf*$^{+/fl}$ controls. (B) Expression of TMEM98 is reduced in the *RPE-Tyrcre-ERT2>Myrf*$^{fl/-}$ mice compared to *Myrf*$^{+/fl}$ controls. (C) Expression of *Myrf* with RNAscope is reduced in the *RPE-Tyrcre-ERT2>Myrf*$^{fl/-}$ mice compared to *Myrf*$^{+/fl}$ controls. The density of

fluorescent staining was quantitated in ImageJ for TMEM98 (D) and *Myrf* (E), normalized to the background staining and expressed as a density per area. Statistics were calculated using the Unpaired T test.
(TIF)

**S2 Fig. Dotplot representation of marker expression used to classify scRNAseq clusters.** Cluster identification is labeled on the left of the graph and the transcript analyzed is listed at the bottom of the graph. Expression is displayed using a gradient where red signifies high levels of expression and blue indicates low levels of expression. The size of the dot is correlated to the number of cells within a particular cluster that expresses the gene.
(TIF)

**S3 Fig. *Pax3* marks the melanocyte cluster and is absent from the RPE cluster.** FeaturePlots showing *Pax3* is specifically expressed in the melanocyte (Mel) cluster and not the RPE cluster, demonstrating a clear distinction between the two clusters.
(TIF)

**S4 Fig. Restriction of TUJ1 staining to the retina in *Rx>cre Myrf*<sup>fl/fl</sup> mutants at P0 supports proper specification of the RPE cells.** TUJ1 staining is in red and auto fluorescent choroidal red blood cells are seen in green. The RPE and retina layers are labeled for orientation. Scale bar represents 50uM.
(TIF)

**S5 Fig. Apoptotic cells detected in *Rx>cre Myrf*<sup>fl/fl</sup> mutants by e15.5.** Cleaved Caspase3 immunostaining was used to detect the presence of apoptotic cells in the RPE of *Rx>cre Myrf*<sup>fl/fl</sup> mutants and controls during late gestation (e15.5) and postnatal timepoints (P0 and P3). Sections through the RPE and retina highlight the pigmentation of the RPE in control mice at e15.5, P0, and P3 (A-C) and the loss of pigmentation beginning at e15.5 in the *Rx>cre Myrf*<sup>fl/fl</sup> mutants (G-I). Apoptosis was analyzed with cleaved-Caspase3 immunostaining. No apoptotic cells were detected in the RPE of control mice across all timepoints analyzed (D-F). Apoptotic cells were detected at e15.5 in *Rx>cre Myrf*<sup>fl/fl</sup> mutant RPE, but not in RPE from postnatal mutants (J-L). Scale bars indicate 50um.
(TIF)

**S6 Fig. Seurat Cell cycle analysis shows similar distribution of cells in G1, G2M, and S phases between *Myrf*<sup>fl/fl</sup> and *Rx>cre Myrf*<sup>fl/fl</sup>.** Seurat cell cycle analysis function was used to analyze changes in cell cycle distribution between *Myrf*<sup>fl/fl</sup> and *Rx>cre Myrf*<sup>fl/fl</sup> scRNAseq datasets. (A). UMAP FeaturePlot of Cell Cycle Scoring split between WT (*Myrf*<sup>fl/fl</sup>) and MUT (*Rx>cre Myrf*<sup>fl/fl</sup>). (B). Pie chart displaying distribution of RPE cells in G1, G2M, and S phases of the cell cycle across each time point and genotype.
(TIF)

**S7 Fig. Pseudotime analysis of reclustered RPE shows a modest shift towards a less differentiated state at P0 in *Rx>cre Myrf*<sup>fl/fl</sup> mutants.** The Monocle3 package in R was used to analyze a pseudotime trajectory across all time points between *Myrf*<sup>fl/fl</sup> and *Rx>cre Myrf*<sup>fl/fl</sup> RPE clustered cells. (A) Seurat reclustered RPE UMAP showing distribution of pseudotime values across genotypes. A pseudotime value of 0 indicates a more progenitor like cell state and values increase as the predicted state of differentiation progresses. (B) Expression of *Mki67* in the Monocle3 RPE cluster was used to determine the location of proliferating cells to use as the beginning of the pseudotime trajectory. (C)(Pseudotime trajectory in the RPE cluster with point 1 designating the most progenitor like state and points 2 and 3 showing two trajectories of increasing pseudotime values coming from the progenitors. (D) Pseudotime density graph across all genotypes highlighting a modest shift in the *Rx>cre Myrf*<sup>fl/fl</sup> towards a less differentiated state at P0.
(TIF)

**S8 Fig. Quantification of ERMN and *Upk3b* expression confirms decrease in *Rx>cre Myrf^{fl/fl}* mutants.** (A) The density of ERMN immunostaining was compared to the density of EZR immunostaining using Image J in both *Myrf^{fl/fl}* controls (n=3) and *Rx>cre Myrf^{fl/fl}* mutants (n=3). Statistical analysis was performed using an unpaired T test. \*\*=p<0.01, \*=p<0.05. (TIF)

**S9 Fig. Loss of *Myrf* does not impact expression of early transcription factors in the RPE.** VlnPlot analysis of *Pax6, Mitf,* and *Otx2* in the RPE cluster, shows comparable levels of expression between mutant and wild type across all time points. (TIF)

**S10 Fig. Quantification of SOX10 positive cells in the RPE confirms loss of expression in the *Rx>cre Myrf^{fl/fl}* mutants.** The total number of SOX10 positive cells were compared to the total DAPI cells in the RPE and graphed as a percentage of SOX10/DAPI. Statistically significant differences were seen between the *Myrf^{fl/fl}* (C), *Rx>cre Myrf^{+/fl}* (H), and *Rx>cre Myrf^{fl/fl}* (M) at all time points. One way ANOVA with Tukey multicomparison was used for statistical analysis. \*\*=p<0.01, \*\*\*=p<0.001, \*\*\*\*=p<0.0001. (TIF)

**S11 Fig. Differential Expression of Prioritized Ligand-receptor pairs.** Ligand-receptor pairs from Differential Nichenetr analysis are prioritized based on log fold change (LFC) expression of the ligand in the RPE. Heat maps show expression of the ligand (left) and the receptor (right) within the RPE of the scRNAseq dataset. (TIF)

**S12 Fig. Quantification of TGFB signaling pathway components.** (A) The density of signal per area from *Wfikkn2* RNAscope was normalized to background staining in ImageJ. (B) The density of signal per area from TGFB2 immunostaining was normalized by comparing to the density of background staining in ImageJ. (C) The density of signal per area from *Id3* RNAscope was normalized by comparing to the density background staining in ImageJ. (D). The density of signal per are of pSMAD immunostaining in the RPE was compared to the density of background staining in ImageJ. (E) VlnPlot of *Bmp2* transcript expression in the RPE cluster of the P0 scRNAseq dataset showing a trend towards increased expression in the mutant. (F) RNAscope of *Bmp2* demonstrates elevated expression in the RPE of *Rx>cre Myrf^{fl/fl}* mutants (n=5) compared to *Myrf^{fl/fl}* controls (n=4). (G). The density of signal per are of *Bmp2* RNAscope in the RPE was compared to the density of background staining in ImageJ. Each point on the graphs represents an individual sample. Ordinary one-way ANOVA with Tukey multicomparisons was used to assess statistics in TGFB2 and *Id3* staining. The unpaired T test was used to assess statistics in *Wfikkn2, Bmp2,* and pSMAD staining. \*=p<0.05, \*\*\*=p<0.001. (TIF)

**S13 Fig. Expression on *MYRF* in human cornea, retina, and RPE.** Expression data was assembled from the Eyeintegration website, provided by the National Eye Institute (eyeintegration.nei.nih.gov). A box plot display of the Pan-Human Gene Expression using the Gene 2019 dataset demonstrates exponentially higher levels (at least 4 log fold) of *MYRF* in the RPE compared to the cornea or retina. (TIF)

**S14 Fig. FeaturePlot analysis of genes altered in reported models of loss of *Myrf*.** FeaturePlot analysis of genes reported to be altered in other models of loss of *Myrf* are not altered in *Rx>cre Myrf^{fl/fl}* mutants. (TIF)

**S1 Table. Genes associated with Gene Ontology (GO) Term enrichment analysis.** (XLSX)

**S2 Table. SCENIC Regulon Activity in the RPE cluster.** Each predicted regulon is followed by the number of genes it is predicted to regulate in parenthesis. (XLSX)

**S3 Table. Genes predicted to be regulated by Sox10.**
(XLSX)

**S4 Table. Pseudobulk RNAseq analysis confirms *Sox10* specifically downregulated in RPE cluster.**
(XLSX)

**S5 Table. Antibody reagents and methods.**
(XLSX)

**S1 Data. Quantification of immunostaining and RNAscope staining.** The first tab in this table shows the density quantifications and cell counts from immunostainings and RNAscopes throughout the paper. Additional tabs in the sheet show the statistical analysis used for each marker.
(XLSX)

**S1 File. Web resources.**
(DOCX)

## Acknowledgments

The authors are grateful to the Advanced Genomics Core for assistance with single cell RNA sequencing; to Athera Yakoo for technical assistance; to Ben Emery for Myrf floxed mice and helpful advice; to Elior Peles for the ERMN antibody; to Kapil Bharti for the *RPE-Tyrcre>ERT2* mice; to Sally Camper, Jun Li, and David Zacks, Brian Brooks, Kapil Bharti, and Robert Hufnagel for helpful discussions.

## Author contributions

**Conceptualization:** Michelle L. Brinkmeier, Lev Prasov.

**Data curation:** Michelle L. Brinkmeier, Su Qing Wang, Hannah A. Pittman, Leonard Y. Cheung.

**Formal analysis:** Michelle L. Brinkmeier, Hannah A. Pittman, Lev Prasov.

**Funding acquisition:** Lev Prasov.

**Investigation:** Michelle L. Brinkmeier, Su Qing Wang, Hannah A. Pittman, Lev Prasov.

**Methodology:** Michelle L. Brinkmeier, Leonard Y. Cheung, Lev Prasov.

**Project administration:** Lev Prasov.

**Software:** Michelle L. Brinkmeier, Leonard Y. Cheung, Lev Prasov.

**Supervision:** Michelle L. Brinkmeier, Lev Prasov.

**Validation:** Michelle L. Brinkmeier, Hannah A. Pittman.

**Visualization:** Michelle L. Brinkmeier.

**Writing – original draft:** Michelle L. Brinkmeier, Lev Prasov.

**Writing – review & editing:** Su Qing Wang, Hannah A. Pittman, Leonard Y. Cheung, Lev Prasov.

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
