## [Decision Letter · Decision Letter 0]

6 Aug 2024

Dear Dr Prasov,

Thank you very much for submitting your Research Article entitled 'Myelin regulatory factor (Myrf) is a critical early regulator of retinal pigment epithelial development.' to PLOS Genetics.

The manuscript was fully evaluated at the editorial level and by independent peer reviewers. The reviewers appreciated the attention to an important problem, but raised some substantial concerns about the current manuscript. Based on the reviews, we will not be able to accept this version of the manuscript, but we would be willing to review a much-revised version. We cannot, of course, promise publication at that time.

If you decide to revise the manuscript for further consideration at PLOS Genetics, please aim to resubmit within the next 60 days, unless it will take extra time to address the concerns of the reviewers, in which case we would appreciate an expected resubmission date by email to plosgenetics@plos.org.

If present, accompanying reviewer attachments are included with this email; please notify the journal office if any appear to be missing. They will also be available for download from the link below. You can use this link to log into the system when you are ready to submit a revised version, having first consulted our Submission Checklist .

PLOS has incorporated Similarity Check , powered by iThenticate, into its journal-wide submission system in order to screen submitted content for originality before publication. Each PLOS journal undertakes screening on a proportion of submitted articles. You will be contacted if needed following the screening process.

To resubmit, log into your Editorial Manager account and select the option 'Revise Submission' in the 'Submissions Needing Revision' folder.

We are sorry that we cannot be more positive about your manuscript at this stage. Please do not hesitate to contact us if you have any concerns or questions.

Yours sincerely,

Seth Blackshaw

Academic Editor

PLOS Genetics

Monica Colaiácovo

Section Editor

PLOS Genetics

Reviewer's Responses to Questions

**Comments to the Authors:**

Reviewer #1: The manuscript by Brinkmeier et al performed scRNAseq analysis of conditional MYRF KO mouse eyes and provided a gene expression and some histological analysis to validate a handful of genes in select pathways. Authors show that MYRF is important for RPE melanogenesis, structure, and survival. Lot of these conclusion were published previously by the same lab. This undermines novelty of the current manuscript. Overall, the work is descriptive with frequent speculative conclusions without supporting evidence.

Specific comments:

1) Authors make claims that MYRF is important for RPE function and EMT induction. But no evidence is provided to support these claims.

2) Authors claim that MYRF is downstream of PAX6 and MITF is not supported by any claim. Authors should analyze MYRF expression in PAX6 and MITF mutant mice.

3) The claim that eye development is normal before E14 needs to be confirmed with appropriate evidence.

4) The claim that SOX10 and MYRF work in the same pathway needs to be substantiated by genetic analysis. Do SOX10 mutant mice phenocopy MYRF mutant mice and the double mutants not have any different phenotype?

5) Authors claim that MYRF expression in the RPE is the prime reason for ocular defects in complete null. Without deleting the ON specific MYRF expression, the previous claim is an overstatement. Can authors rule out the possibility that ON-specific MYRF expression does not result is secretion of certain growth factors that regulate RPE and eye development.

6) Clearly MYRF deletion in the RPE result in loss of several hundred genes including critical RPE genes that have been associated with AMD. Without a direct association of MYRF with AMD, this statement is highly speculative. MYRF is a developmental transcription factor and AMD is an age-onset disease.

7) Regulon analysis provided in Figure 8 is interesting by highly speculative and, as authors say, ‘predictive’. It is not a confirmatory proof and needs to be substantiated with additional genetic, epigenomic, and in vitro analysis. No methods are provided for this figure.

8) Lines 160-161, how come a 2% reduction in RPE cell number result in in a 8% increase in total RPC number. What is the total number of cells in each case? Are these numbers statistically significant? If yes, are authors suggesting RPE transdifferentiated into retina. If that’s the case, they should provide histological and gene expression evidence in support of that. If not, this data should be removed.

9) Lines 177-179, If MYRF is expressed in all of RPE cells, why do only a small number of cells die?

10) Lines 185-187 contradict data shown in lines 160-161.

11) Lines 235-237, how do authors conclude depigmentation is due to loss of melanosomes. Can authors rule out slow proliferation of some RPE cells as others are dying? This also doesn’t rule out reduced melanin production. A longitudinal analysis from before the KO and after the KO will help make this claim.

12) Lines 306-397, why is elevated TGF-beta signaling seen in RPCs? Is this coming from RPE cells and what is the impact of this on RPCs. It seems their proliferation and differentiation is not impacted.

13) Lines 349-350, no evidence is provided for correct RPE specification.

14) Lines 366-367, no data is provided for loss of RPE hexagonal feature.

15) Lines 398-299, has RPE cell death been reported in MITF KO mice. Don’t these mice have more of a RPE specification defect? Please clarify or correct.

16) What is the evidence and consequence for increased EMT gene expression?

Reviewer #2: The transcription factor Myrf is critical during eye development in humans; genetic variants can cause nanophthalmos but the underlying functional role for MYRF is very poorly understood. Using wildtype and conditional Myrf mutant mice, this study employed sc RNAseq of mouse optic cups containing retina, RPE and neighboring cell types collected at 3 different time points between E13.5 and birth. One cluster harboring RPE cells together with melanocytes was identified, which was further examined thoroughly with various computational analyses to identify downstream candidates critical for RPE development. This work has great potential to represent an important step toward identifying critical Myrf targets and for elucidating the role of Myrf specifically in RPE and ocular development.

There are 2 major concerns:

1) It appears that the initial clustering yielded a mixed RPE and melanocyte cluster (RPE/Mel), since the identification genes are also expressed in melanocytes (Suppl Fig.1). Is it possible that a rather significant number of cells in this cluster are melanocytes? For example, Fig.2B shows that there are many MYRF-negative cells in the 'RPE' cluster (= RPE/Mel cluster?), however most of the RPE cells in Fig1A (E14.5) and B (P0) are MYRF-positive. The RPE cluster may be able to be separated from melanocytes as a subcluster? For example, Otx2 and Lhx2 are expressed early on and in the postnatal RPE, also shown in Suppl Fig.1. Pax6 is another marker that can distinguish RPE and melanocytes. Also, while the authors define the 'RPE/Mel' cluster in Suppl. Fig.1, they use throughout the manuscript just the term 'RPE' cluster, which is confusing.

2) In many figures, colocalization with relevant genes in RPE or other structures such as optic nerve, melanocytes or appropriate higher magnifications are missing, among them Fig.1, Fig.6 (incl. a previously unknown gene expressed in RPE UpK3b), Fig.7. This applies particularly to Fig.9, the magnification is too low to confirm whether the labeling of TGF pathway genes is present in RPE cells. This is important for validation of the scRNAseq data, especially if this a mixed RPE/melanocyte cluster. There is also a concern that very few cells in the RPE in Myrf mutants show upregulated pSMAD labeling, and the effect on TGFB2 and BMP2 is not very clear. And what does the green labeling in Fig.9E in wildtype tissue represent? Fig.7: Sox10 expression appears to variable at E13.5, compared to expression in melanocytes, and very sparse at P0. Also here, it would be helpful to see colocalization with RPE marker.

Minor points

- Rx-Cre can also be expressed in the lens. Please, provide information whether MYRF is expressed in lens.

- Is there anything known about Sox10 function in the developing eye?

- How is "highest intensity Otx2 stained" or 'less intensity stained' defined?

Reviewer #3: In this study, Brinkmeirer, Prasov and colleagues investigate the role of MYRF in RPE cells using conditional deletion in mice and employing single cell RNAseq analysis. The data presented, as well as the analysis is of high quality and well controlled. I do feel the study in its current form falls a bit short of PlosGen standards, though. Some follow-up experiments could add the needed depth. The authors go to great length arguing an RPE-centric roll for Myrf function on eye size control and PhR health. But the critical experiment of deleting myrf only within RPE is missing. Whole optic cup deletion with rx:cre leaves the window open for other interpretations. Even though myrf is more highly expressed in the RPE as compared to other parts of the eye, the protein could still have supplementary and significant roles outside of the RPE. Some follow-up experimentation with the other major conclusions is also warranted. First, Sox10 is implied as a direct target of Myrf. This should be evaluated. Last, it is observed that BMP/TGFb signaling is enhanced in myrf cKO eyes. What is the basic mechanism underlying this effect? Any does this pathway activation drive subsequent phenotypes. Yes, literature supports this possibility, but I think experimental tests should be conducted.

In short, this is a very solid start to a potentially compelling study. My recommendation is the push the experimentation just a little further to really enhance the impact of this nice research.

**Have all data underlying the figures and results presented in the manuscript been provided?**

Reviewer #1: Yes

Reviewer #2: Yes

Reviewer #3: Yes

PLOS authors have the option to publish the peer review history of their article (what does this mean? ). If published, this will include your full peer review and any attached files.

**Do you want your identity to be public for this peer review?** For information about this choice, including consent withdrawal, please see our Privacy Policy .

Reviewer #1: **Yes: ** Kapil Bharti

Reviewer #2: No

Reviewer #3: No

---

## [Decision Letter · Decision Letter 1]

22 Jan 2025

PGENETICS-D-24-00656R1

Myelin regulatory factor (Myrf) is a critical early regulator of retinal pigment epithelial development.

PLOS Genetics

Dear Dr. Prasov,

Thank you for submitting your manuscript to PLOS Genetics. After careful consideration, we feel that it has merit but does not fully meet PLOS Genetics's publication criteria as it currently stands. Therefore, we invite you to submit a revised version of the manuscript that addresses the points raised during the review process.

Please submit your revised manuscript within 60 days Mar 23 2025 11:59PM. If you will need more time than this to complete your revisions, please reply to this message or contact the journal office at plosgenetics@plos.org. Please include the following items when submitting your revised manuscript:

We look forward to receiving your revised manuscript.

Kind regards,

Seth Blackshaw

Academic Editor

PLOS Genetics

Monica Colaiácovo

Section Editor

PLOS Genetics

Aimée Dudley

Editor-in-Chief

PLOS Genetics

Anne Goriely

Editor-in-Chief

PLOS Genetics

**Journal Requirements:**

1) We have noticed that you have uploaded Supporting Information files, but you have not included a complete list of legends. Please add a full list of legends for the supplementary tables after the references list.

2) In the online submission form, you indicated that the "primary data are available upon request from the corresponding author."  All PLOS journals now require all data underlying the findings described in their manuscript to be freely available to other researchers, either

1. In a public repository

2. Within the manuscript itself

3. Uploaded as supplementary information.

3) We note that the author Lev Prasov does not have an affiliation listed in the online submission form. Please ensure that the affiliation is added in the online submission form.

4) Please ensure that the funders and grant numbers match between the Financial Disclosure field and the Funding Information tab in your submission form. Note that the funders must be provided in the same order in both places as well. Currently, the order of this grant "K12EY022299" is different in both places. In addition, the grant number provided by the National Eye Institute does not match in the Financial Disclosure field and the Funding Information tab.

Please indicate by return email the full and correct funding information for your study and confirm the order in which funding contributions should appear. 

**Reviewers' comments:**

Reviewer's Responses to Questions

Reviewer #1: none

Reviewer #2: The revisions do not address satisfactorily many of the concerns.

In the revised version, the authors provided higher magnifications of the presumed RPE region in insets in several figures. However, this does not resolve my concern whether gene expression is localized to RPE. One way to address this is trying RNAscope double-labeling with 2 different probes or combining RNAscope with immunohistochemistry. Also, if double-labeling with RPE markers is not possible with the validated Myrf antibody that the authors used in their previous study (Garnai et al., 2019), the current Myrf transcript labeling should be at least overlaid with brightfield at high magnification to show overlap with RPE pigmentation in controls and also in Myrf CKO, since some pigmentation is still present. Also, were the images processed the same way?

The magnification in Fig.1K is too low, and the orientation in L is unclear. Is the lens in Fig.1L shown as a whole? Overall, with these corrections, this figure would then also more convincingly confirm their previous study from 2019. Similarly, in Fig. 6, RPE expression of Upk3b would be novel, however this needs to be shown with an RPE marker colocalized. In Fig.7, it appears that Sox10 is only weakly expressed in conditional hets at E13.5? In general, changes in RPE development can affect differentiation of extraocular tissues, thus, it should be confirmed that Sox10 is expressed in melanocytes. In Fig.9, the effects on TGFb2, BMP2, Id3 and pSmad expression in CKO RPE are not convincingly shown, whether it is scRNAseq data (Fig.9A) or RNAscope or IHC for pSMAD (C-F). This need to be quantified and colocalization with RPE markers at higher magnification needs to be shown. Overall, the key conclusions for this study are not supported by the data presented here.

Comments regarding new data:

Fig. S1: The data obtained for disrupting Myrf in the developing RPE using Tyrp-Cre-ERT2 is not convincing. The occurrence of Myrf transcript (red label) in the region of the RPE in the CKO does not really look different from control. Quantification should be provided. And where would the choroid in all samples be located? How many animals per genotype were analyzed? Labels for tissue types, lines and arrows would be very helpful.

Additional comments:

The manuscript needs thorough checking whether revisions have been completed for every section (including methods), and whether subtitles need to be adjusted. For example, the subtitle in line 144 is not accurate anymore, since only one population (RPE) appears altered in Myrf CKO.

Fig.3: TUNEL

The units for the y-axes should be added. T-test is not appropriate if 3 groups are compared. It should be mentioned whether both eyes for each animal were analyzed. Line 82: The word "controls" is in the wrong place.

Overall, the scRNAseq data is described in a fine way and points to potentially interesting mechanisms of Myrf's role during RPE differentiation. It would be significantly enhanced, if causative relationships between Myrf and downstream factors (Sox10, TGFB/BMP) are mechanistically investigated.

Reviewer #3: While additional analysis would strengthen this work, my main concern about RPE autonomy was addressed.

**Have all data underlying the figures and results presented in the manuscript been provided?**

Reviewer #1: Yes

Reviewer #2: Yes

Reviewer #3: Yes

PLOS authors have the option to publish the peer review history of their article (what does this mean? ). If published, this will include your full peer review and any attached files.

**Do you want your identity to be public for this peer review?** For information about this choice, including consent withdrawal, please see our Privacy Policy .

Reviewer #1: No

Reviewer #2: No

Reviewer #3: No

**Figure resubmission:**
---

## [Editor Report · Decision Letter 2]

1 Apr 2025

Dear Dr Prasov,

We are pleased to inform you that your manuscript entitled "Myelin regulatory factor (Myrf) is a critical early regulator of retinal pigment epithelial development." has been editorially accepted for publication in PLOS Genetics. Congratulations!

Yours sincerely,

Seth Blackshaw

Academic Editor

PLOS Genetics

Monica Colaiácovo

Section Editor

PLOS Genetics

Aimée Dudley

Editor-in-Chief

PLOS Genetics

Anne Goriely

Editor-in-Chief

PLOS Genetics

Comments from the reviewers (if applicable):

**Data Deposition**

http://datadryad.org/submit?journalID=pgenetics&manu=PGENETICS-D-24-00656R2

**Press Queries**

---

## [Editor Report · Acceptance letter]

PGENETICS-D-24-00656R2

Myelin regulatory factor (MYRF) is a critical early regulator of retinal pigment epithelial development.

Dear Dr Prasov,

We are pleased to inform you that your manuscript entitled "Myelin regulatory factor (MYRF) is a critical early regulator of retinal pigment epithelial development." has been formally accepted for publication in PLOS Genetics! Your manuscript is now with our production department and you will be notified of the publication date in due course.

With kind regards,

Zsofia Freund

PLOS Genetics

On behalf of:
